# Parathyroid Hormone as a Modulator of Skeletal Muscle: Insights into Bone–Muscle and Nerve–Muscle Interactions

**DOI:** 10.3390/ijms26157060

**Published:** 2025-07-22

**Authors:** Vinh-Lac Nguyen, Kwang-Bok Lee, Young Jae Moon

**Affiliations:** 1Department of Orthopaedic Surgery, Jeonbuk National University Medical School, Jeonju 54896, Republic of Korea; nvlac@huemed-univ.edu.vn (V.-L.N.); osdr2815@naver.com (K.-B.L.); 2Department of Biochemistry and Molecular Biology, Jeonbuk National University Medical School, Jeonju 54896, Republic of Korea; 3Department of Surgery, Hue University of Medicine and Pharmacy, Hue University, Hue 530000, Vietnam; 4Department of Orthopaedic Surgery, Jeonbuk National University Hospital, Jeonju 54896, Republic of Korea; 5Biomedical Research Institute, Jeonbuk National University Hospital, Institute for Medical Sciences and Research Institute of Clinical Medicine, Jeonbuk National University, Jeonju 54896, Republic of Korea

**Keywords:** parathyroid hormone, PTH, skeletal muscle, bone–muscle, nerve–muscle

## Abstract

Parathyroid hormone (PTH) has been studied to determine its broader role in musculoskeletal health, particularly its effects on skeletal muscle. Bone and muscle are inextricably linked via mechanical loading and biochemical signaling, with both processes playing important roles in muscular metabolism and function. Furthermore, the nervous system must maintain muscle mass and function, as neuromuscular transmission controls muscle contraction, protein synthesis, and energy metabolism. As a systemic endocrine regulator, PTH influences the physiology of skeletal muscle—both directly and through interactions with bone and the nervous system, modulating myokines, osteokines, and neuromuscular activity. The intricate relationships between PTH, muscle, bone, and nerves continue to be investigated due to their implications for aging, metabolic pathologies, and musculoskeletal disorders.

## 1. Introduction

Parathyroid hormone (PTH) is a critical endocrine regulator of calcium and phosphate homeostasis, primarily known for its effects on bone and kidney metabolism [1]. Recent studies have highlighted its broader role in musculoskeletal health, particularly its effects on skeletal muscle. Meanwhile, bone and muscle are closely interconnected through both mechanical loading and biochemical signaling, playing key roles in muscle metabolism and function [2,3]. Furthermore, the nervous system is essential for maintaining muscle mass and function as neuromuscular signaling regulates muscle contraction, protein synthesis, and energy metabolism through calcium control [4,5]. Considering the classical role of PTH and the muscle symptoms (muscle weakness, myopathy, etc.) that occur in hyperparathyroidism and hypoparathyroidism, it is thought that PTH may affect these musculoskeletal and neuromuscular interactions in a complex manner [6]; however, there is a lack of comprehensive information on the systemic and local effects of PTH on skeletal muscle. Investigating the molecular pathways involved in PTH-mediated muscle regulation is expected to provide valuable insights into musculoskeletal diseases—particularly those related to aging, metabolic disorders, and endocrine dysfunction—and to suggest targeted treatment strategies for various musculoskeletal conditions.

Our review synthesizes emerging evidence on the non-classical roles of PTH in skeletal muscle and neuromuscular junction function. By integrating findings from clinical, in vivo, and in vitro studies, we provide a novel perspective on PTH as a potential therapeutic modulator in the muscle–bone–nerve crosstalk, addressing a critical gap in current musculoskeletal research.

## 2. Methodology

A comprehensive literature search was conducted to identify relevant studies published up to 2025, with a focus on the role of parathyroid hormone (PTH) in skeletal muscle and its interactions with bone and nerve systems. The primary database used was PubMed, supplemented by Scopus, Web of Science, ScienceDirect, and Embase to ensure broad coverage of biomedical and interdisciplinary research.

The following keywords and their combinations were used in the search: parathyroid hormone, PTH, skeletal muscle, hormones, bone–muscle, nerve–muscle, myokines, osteokines, neuromuscular junction, and NMJ. In addition, we included the names of specific myokines and osteokines (e.g., IL-6, irisin, FGF23, and osteocalcin) and NMJ components (e.g., acetylcholine cluster, Agrin, and MuSK) to refine and deepen the search. Operators such as “and” and “to” were applied to structure complex queries and optimize retrieval.

The inclusion criteria were as follows:-Original research articles, reviews, and meta-analyses published in English.-Studies addressing the role or effects of PTH (or its analogs) on skeletal muscle, neuromuscular junctions, or crosstalk between muscle and bone or nerve.-Studies involving in vivo, in vitro, or clinical investigations.

The exclusion criteria were non-peer-reviewed sources, editorials, and conference abstracts without primary data.

To ensure the depth and accuracy of the review, reference lists of key publications were also manually examined to identify additional pertinent studies. All retrieved records were screened based on title and abstract, and a full-text review was conducted to confirm eligibility.

For example, to explore the effects of PTH on a myokine such as irisin and its target organ, skeletal muscle, we searched PubMed with the keywords “PTH and irisin and muscle” and found seven results. Next, the keywords “irisin and muscle and bone” or “irisin and muscle and nerve” were searched to learn more about how this myokine affects these organs. The results were filtered to be relevant to the topic of our review. We repeated this process for the other databases. The most recent findings or reviews were prioritized to avoid duplicating earlier studies and to reflect the latest advancements.

## 3. Parathyroid Hormone

PTH is the product of the parathyroid glands, which are nodular structures usually located along the dorsal part of the thyroid [7]. PTH, calcitriol (1,25-dihydroxyvitamin D), and fibroblast growth factor 23 (FGF23) are three key hormones modulating calcium and phosphate homeostasis. PTH synthesis occurs within the parathyroid glands, starting with a 115-amino-acid polypeptide; it then produces the primary hormone, active 84-amino-acid PTH, which is stored, secreted, and functions in the body [1]. All of the known biological actions operate within the 34 residues of its NH2-terminal (PTH 1–34) [1,8].

PTH receptors are classified as PTH1R, PTH2R, or non-classical receptors. Research has revealed the expression of PTH1R in skeletal muscle, including satellite cells [9]. Parathyroid hormone-related protein (PTHrP) exhibits similarities to PTH, frequently acting in a local paracrine or autocrine manner [10]. Both PTH and PTHrP can be recognized by PTH1R because of the high degree of similarity in the amino-terminal regions of these two peptides [9,11].

PTH acts differently on the neuromuscular system depending on the degree of exposure. Intermittent PTH exposure, such as in osteoporosis treatments, has been shown to have anabolic effects on bone, which might indirectly benefit muscle by enhancing bone-derived signaling molecules and providing mechanical support [12]. Additionally, PTH can modulate neuromuscular function, potentially influencing the nerve conduction and synaptic activity essential for muscle contraction and coordination [13,14]. However, chronic high PTH levels—as seen in conditions such as hyperparathyroidism and chronic kidney disease—are associated with muscle weakness, increased protein degradation, and impaired muscle cell regeneration [6,15].

To the best of our knowledge, the United States Food and Drug Administration (FDA) has approved three PTH-based medicines for clinical use. Forteo (teriparatide), a recombinant PTH (1–34) fragment, is authorized for the treatment of osteoporosis in postmenopausal women and men at high risk for fractures. Tymlos (abaloparatide), a synthetic analog of PTH-related peptide (PTHrPs 1–34), is also indicated for the treatment of osteoporosis. Yorvipath (palopegteriparatide), approved in 2024, is a prodrug of PTH (1–34) intended to provide continuous PTH exposure for adults with hypoparathyroidism. These medicines are critical for treating bone and mineral metabolism problems, yet their emerging roles in skeletal muscle and neuromuscular health are not fully understood. These agents may exert direct or indirect effects on muscle regeneration and neuromuscular junction stability, representing a promising but mechanistically underexplored therapeutic avenue.

## 4. Bone and Skeletal Muscle Interactions

One of the largest systems in the human body is the musculoskeletal system, which principally comprises bone, articular cartilage, tendon, ligament, and muscle tissues [16]. During embryogenesis, myogenesis occurs in close proximity to and simultaneously with skeleton formation [17]. Bone and muscle mass increase significantly and proportionally during postnatal growth, reaching their peak mass at about the same time [18,19]. Aging and injury can cause bone and muscle degradation and dysfunction at the same time; as such, therapies that target pathways that centrally regulate both these two tissues might provide a significant benefit [20].

The concepts of a “bone–muscle unit” and a “muscle–bone unit” have appeared in bone and skeletal muscle research studies to emphasize the crosstalk between these two major components of the musculoskeletal system [3,21]. Indeed, from the cellular and tissue levels up to the organismic and systemic levels, their communication shows up in multiple aspects: diversity among the physical forces produced by gravity and the outside environment; impacts from each other’s secretory products; and systemic endocrine and other effector interactions [2,3,22]. The bidirectional effects of myokines and osteokines on bone and muscle mentioned below are summarized in Table 1 and Figure 1.

### 4.1. Bone to Muscle

#### 4.1.1. FGF23

Fibroblast growth factors (FGFs) are secreted proteins essential for regulating cell division, migration, survival, and proliferation throughout embryonic and postnatal development [23,24]. FGF23 controls phosphate homeostasis, lowers vitamin D levels, and is produced primarily by mature osteoblasts and osteocytes [25]. FGF23 overexpression leads to hypophosphatemia and skeletal abnormalities [26]. FGF23 reduces bone marrow mesenchymal stem cell osteogenesis via the FGFR3-ERK signaling pathway and suppresses mature osteoblast mineralization [27]. A high FGF23 serum level is reportedly a reliable predictor for poor bone mass postmenopause [28].

FGF23 regulates α-klotho expression through a novel bone-kidney axis [29]. A soluble co-receptor of FGF23 in plasma, klotho prevents the development of cultured skeletal muscle cells by suppressing insulin/IGF-1 signaling [30]. Furthermore, FGF23 and klotho affect smooth muscle and myocardial tissue [31,32]. A comprehensive review demonstrated that FGF23 is linked to a variety of skeletal muscle-wasting processes in chronic kidney disease, including insulin resistance, changes in adipocytokine metabolism, increased oxidative stress, and local and systemic inflammation [33].

#### 4.1.2. PGE2

Prostaglandin E2 (PGE2) is a multifunctional molecule whose production is controlled by various factors, mainly cyclooxygenase 2. This proinflammatory mediator is produced in bone (primarily by osteoblasts) and can regulate both bone resorption and formation processes [34,35]. PGE2 could be involved in the Wnt pathway through its pivotal role in bone; for example, PGE2 can trigger β-catenin signaling [36]. Furthermore, PGE2 has been reported to play a central role in the cAMP/PKA signaling pathway [37].

Myogenesis can be regulated by PGE2 signaling through EP4 receptor activation, which promotes skeletal muscle myoblast proliferation [38]. In addition, PGE2 is vital for the function of muscle-specific stem cells, contributing to muscle regeneration and strength [39].

#### 4.1.3. Osteocalcin

Osteoblasts primarily secrete osteocalcin (or bone γ-carboxyglutamic acid protein), one of the most abundant proteins in bone; it is a vitamin K-dependent, non-collagenous protein [40,41]. Osteocalcin has recently been identified as a bone-derived factor that affects glucose metabolism, reproduction, and cognition via endocrine loops between bone and the pancreas, testes, and brain [41]. In a recent investigation, Wang et al. presented evidence that undercarboxylated osteocalcin had unfavorable effects on the early differentiation of osteoclasts and consequent bone resorption via GPRC6A (G protein-coupled receptor class C group 6) [42].

Osteocalcin directly elevates glucose transport in adipose tissues and muscle cells [43]. Among other things, osteocalcin elevates the muscle uptake and use of glucose and fatty acids and promotes interleukin-6 (IL-6) myokine production during exercise [44,45].

#### 4.1.4. IGF-1

Insulin-like growth factor-1 is produced in the liver, skeletal muscles, and many other tissues in response to growth hormone stimulation, helping to promote normal bone and tissue growth and development. PI3K/Akt and MAPK/ERK are well known pathways through which IGF-1 stimulates the IGF-1 receptor [46]. IGF-1 can mediate most GH effects and thereby directly influence skeletal cells. The differentiation of osteoblasts and bone production is improved by IGF-1 [47]. It elevates skeletal muscle protein synthesis, inhibits muscle atrophy, and promotes skeletal muscle regeneration [48].

#### 4.1.5. Sclerostin

Osteocytes produce a secreted glycoprotein called sclerostin, a product of the SOST gene. Sclerostin antagonizes Wnt signaling by binding to low-density lipoprotein receptor-related proteins (LRPs) 5 and 6 [49] and is a negative regulator of bone formation [50,51]. In patients with osteoporosis, sclerostin is a candidate biomarker due to its association with lower bone mineral content and bone density [52]; therefore, a sclerostin inhibitor could be a promising therapy for bone-related disorders.

As noted, sclerostin interacts with and inhibits LRPs, playing a vital part in the Wnt signaling pathway that has been shown to affect muscle [53]. Furthermore, sclerostin inhibition ameliorates bone metastases and muscle weakness in breast cancer patients with musculoskeletal complications [54].

### 4.2. Muscle to Bone

#### 4.2.1. Myostatin

Myostatin, or growth differentiation factor-8, is an autocrine and paracrine hormone in the transforming growth factor-beta (TGF-β) superfamily that is generated by muscle cells and suppresses muscle differentiation and growth [55]. Although the possible advantage of myostatin insufficiency in muscular function and energetics is still debated [56], many published studies describe how a lack or inhibition of myostatin significantly increases skeletal muscle mass [55].

Researchers have recently proposed that myostatin is an undesirable modulator of bone formation and metabolism [57]. Osteoblastic differentiation is directly inhibited by myostatin via a mechanism involving osteocyte-derived exosomal miR-218 [58]. Additionally, myostatin directly regulates differentiation in osteoclasts [59].

Several studies have revealed vital pathways and modulators that could be related to myostatin: ERK1/2 cascade, Wnt pathways, TGF-β1, and IGF-1 [55].

#### 4.2.2. IGF-1

Muscle fibers can secrete IGF-1 [60]. The functions of this myokine on bone and skeletal muscle are described in the previous section.

#### 4.2.3. Osteoactivin

A type I transmembrane glycoprotein and a rat homolog of nonmetastatic melanoma protein B called *osteoactivin* is highly expressed in human melanoma cells [61]. The expression of osteoactivin in skeletal muscle is elevated in response to spaceflight, tail suspension, or denervation [62], and it is believed to protect injured muscle from fibrosis, resulting in full regeneration after denervation [63]. There have been discussions about the effects of osteoactivin on bone in both directions. Prior studies indicate that osteoactivin might bind to αvβ3 integrin and induce bone resorption [64,65]. In contrast, osteoblast differentiation in vitro and bone formation in vivo are reduced by osteoactivin mutation [66]. Additionally, osteoactivin is a favorable modulator of bone development both in vitro and in vivo [61].

#### 4.2.4. Interleukins

During exercise, muscle secretes IL-6, an inflammatory factor with major effects on the liver, adipose depots, and bone, among other tissues [67,68]. In a comprehensive review by the Muñoz-Cánoves team, IL-6 appeared to have bidirectional effects on skeletal muscle. IL-6 benefits muscle formation, growth, and regeneration, and it can modulate satellite cell-dependent myogenesis. In contrast, evidence also shows that IL-6 increases the velocity of muscle protein synthesis and breakdown, and IL-6 signaling is engaged during muscle atrophy [69]. Similarly, IL-6 shows up as a double-edged sword in bone tissue despite its important role. While it promotes bone growth, it can also cause bone loss in several osteolytic diseases [70,71].

IL-7 might affect satellite cells during muscle development, and is secreted and expressed by human skeletal muscle cells [72,73]. This powerful lymphopoietic cytokine is reported to cause bone loss and have a bone-wasting effect [74]. Moreover, the Weitzmann team revealed that IL-7 can target both the osteoclast and osteoblast pathways, triggering bone resorption and inhibiting bone formation [75].

Interleukin-15 (IL-15), a member of the four α-helix bundle family of cytokines, is generated by a range of tissues and highly expressed in skeletal muscle in response to exercise [76,77]. IL-15 is a possible target for preserving healthy skeletal muscle and treating muscle-wasting syndromes through multiple actions, including the stimulation of myoblast development and skeletal muscle fiber growth and the prevention of protein breakdown [78]. The role of IL-15 in bone metabolism is complex, as shown by conflicting results from various research studies. Depending on the physiological or pathological state, IL-15 might have a bidirectional regulatory function that produces varying effects on osteoblast or osteoclast proliferation and differentiation, skeletal matrix deposition, and the resorption of injured bone tissue [79].

#### 4.2.5. Irisin

A novel myokine called irisin is released subsequent to the proteolytic cleavage of its precursor, fibronectin type III domain-containing 5 (FNDC5), and participates in energy metabolism, aging, inflammation, and neurogenesis [80,81]. In skeletal muscle, irisin induces mitochondrial biogenesis [82] and upregulates myogenesis by increasing myogenin, the myonectin transcript level, and myocyte cell proliferation and decreasing the mRNA expression of myostatin, dystrophin, MuRF1, and MAFbx [83]. Colaianni claimed that irisin protected against muscle atrophy and bone loss in an in vivo experiment [84].

Recent evidence has demonstrated that irisin regulates bone homeostasis through its interactions with both osteogenesis and osteoclastogenesis [85,86]. Irisin has been found to affect the αV integrin-induced ERK/STAT pathway and BMP/SMAD signaling activation in osteogenesis promotion [87]. Irisin also takes part in the Wnt/β-catenin signal pathway, upregulates autophagy, and increases the osteogenic differentiation of bone marrow mesenchymal stem cells [88]. Moreover, in a mouse model of osteogenesis imperfecta, irisin was found to antagonize TGF-β/Smad signaling and assist osteogenesis, reducing bone fractures [89].

#### 4.2.6. FGF2

FGF2 production occurs in the vascular smooth muscle cells and the distal nephron [90], enhancing aged skeletal muscle by stimulating muscle growth and promoting intramuscular adipogenesis [91]. Notably, this muscle output affects bone growth and cartilage repair [92,93,94]. A combination of FGF2 and BMP-2 interval intramuscular injections significantly increases the osteogenic potential of mesenchymal stem cells and the mRNA expression of an osteoblast gene marker; furthermore, the FGF2 signaling pathway and BMP-2 elevate Runx2 expression during the phosphorylation of ERK/Runx2 [95].

#### 4.2.7. Musclin

Musclin is a bioactive protein produced and secreted by skeletal muscle and plays a physiological role in glucose metabolism [96,97]. By promoting mitochondrial biogenesis, musclin can boost physical endurance [98]. It might also affect bone by repressing FOXO1 [99], an important regulator that affects bone resorption [100].

**Table 1 ijms-26-07060-t001:** Bone and skeletal muscle interactions.

Osteokines/Myokines	Effects on Bone	Mechanism/Signaling Involved	Effects on Skeletal Muscle	Mechanism/Signaling Involved
FGF23	In vitro: ↓ BMSCs osteogenesis, ↓ mature osteoblast mineralization [27]	FGFR3-ERK [27]	In vitro: ↓ Muscle cell differentiation [30]	Insulin/IGF-1, klotho [30]
	Clinical: Serum level associated with low bone mass [28]		Relevant to skeletal muscle wasting [33]	
PGE2	Regulating both bone resorption and formation processes [34,35]		In vitro: Myogenesis [38]	EP4 receptor [38]
			In vivo, in vitro: Muscle regeneration and strength [39]	Muscle-specific stem cells [39]
	Related: Wnt, β-catenin [36], cAMP/PKA [37]			
Osteocalcin	Glucose metabolism, reproduction, and cognition [41]		In vitro: ↑ Glucose transport [43] Clinical, in vivo: ↑ Muscle uptake [44,45]	
	In vitro: ↓ Osteoclasts differentiation [42]	GPRC6A [42]		
IGF-1	↑ Osteoblasts differentiation, bone production [47]		↑ Protein synthesis and regeneration ↓ Muscle atrophy [48]	
	Related: PI3K/Akt, MAPK/ERK [46]			
Sclerostin	Clinical, in vivo, in vitro: ↓ Bone formation [50,51]	Wnt [49]	Clinical: ↑ Muscle weakness [54]	
Myostatin	↓ Bone formation Metabolism [57]		↓ Muscle mass [55]	
	In vitro: ↓ Osteoblastic differentiation [58]	Osteocyte-derived exosomal miR-218 [58]		
	↓ Osteoclast differentiation [59]	RANKL, NFATC1 [59]		
	Related: ERK1/2, Wnt, TGF-β1, IGF-1 [55]			
Osteoactivin	In vivo, in vitro: ↓ Osteoclastogenesis [61]	CD44-ERK [61]	In vivo: Protection from fibrosis [63]	MMP-3, MMP-9 [63]
	In vivo, in vitro: ↑ Bone formation [66]	TGF-β [66]		
IL-6	↑ Bone growth [70,71]		↑ Formation, growth, regeneration, satellite-cell-dependent myogenesis [69]	
	↑ Bone loss (in several osteolytic diseases) [70,71]		↑ Protein synthesis and breakdown Engaged with muscle atrophy [69]	
IL-7	In vivo: ↑ Bone loss [74]	RANKL	Might affect satellite cells [72,73]	
IL-15	Bidirectional regulatory function [79]		Clinical: ↑ Myoblast development, fiber growth; ↓ protein breakdown [78]	
Irisin	In vivo: ↓ Bone loss [84]		In vitro: Mitochondrial biogenesis [82] In vivo: ↑ Myogenesis [83] ↓ Muscle atrophy [84]	
	Related: MAPK [85], ERK/STAT, BMP/SMAD [87], Wnt/β-catenin [88]			
FGF2	Bone growth [92,93,94]		Clinical, in vivo: ↑ Muscle growth, intramuscular adipogenesis [91]	miR-29a/SPARC [91]
	In vivo: Bone marrow MSC Osteogenesis [95]	ERK/Runx2 [95]		
Musclin	Bone resorption [100]	RANKL [100]	Glucose metabolism [96,97]	
			In vivo: ↑ Physical endurance [98]	Mitochondrial biogenesis

BMSCs, bone marrow mesenchymal stem cells; GPRC6A, G protein-coupled receptor class C group 6; RANKL, receptor activator of nuclear factor κB ligand; NFATC1, nuclear factor of activated T-cells; MMP, matrix metalloprotease; MSCs, mesenchymal stem cells.

## 5. Muscle and Nerve Communication

Skeletal muscles must be stimulated by a motor neuron to contract [101]. A *Motor unit* is defined as a single anterior horn cell, its motor neuron, all the skeletal muscle cells stimulated by that neuron, and their neuromuscular junctions [4]. The all-or-none law states that the strength of a nerve cell or muscle fiber’s response does not depend on the strength of the stimulus. If a stimulus is above a certain threshold, a nerve or muscle fiber will fire. Essentially, there will either be a full response or there will be no response at all [102,103]. The interactions between nerves and muscles are detailed in Table 2 and Figure 2.

### 5.1. Neuromuscular Junctions

Neuromuscular junctions (NMJs), first described in the 1840s, are regions where motor neurons communicate with muscle fibers [104]. Different theories about the contact between neurons and the sarcolemma were intensely debated until the middle of the 20th century [105], but advances in electron microscopy ended these discussions by enabling the three-dimensional modeling of the NMJ ultrastructure and the first descriptions of synaptic clefts [106].

The NMJ structure has three main parts: the nerve terminal (the presynaptic part), the motor endplate (the postsynaptic part), and the synaptic cleft (the area between the nerve terminal and motor endplate). Motor neurons lose their myelin sheath when they reach the target muscle, developing a complex of 100–200 branching nerve endings called nerve terminals that comprise active zones (a family of SNAP proteins and rows of voltage-gated Ca channels), potassium channels, endoplasmic reticulum, mitochondria, and synaptic vesicles. Each NMJ contains 5000–10,000 molecules of the NMJ neurotransmitter acetylcholine (ACh). The synaptic cleft is approximately 50 nm and contains the acetylcholinesterase enzyme, and it is the space where ACh is released to engage with nicotinic ACh receptors on the motor endplate [107,108,109].

The discovery of many important components of the NMJ, including agrin, rapsyn, MuSK, and LRP4 [110,111,112,113,114,115], has produced a large field of research studying their interactions and downstream mechanisms in NMJ formation, development, maintenance, and aging.

### 5.2. Nerve to Muscle

The membrane and contractile characteristics of muscles are determined by motor neurons located in the brainstem and spinal cord. If a muscle’s nerve is cut, the muscle loses its function. After disconnection, the consequences of denervation occur, such as changes in gene expression in the muscle fibers, fibrillations, and acetylcholine supersensitivity. However, if there is a re-innervative condition, the muscle fibers will have new phenotypes that correspond to their new regenerating axons instead of the old ones [116,117]. An in vitro study showed that electrical and metabolic methods can induce pre-fusing myoblasts or myoblast-myotubes in gap junctions [118]. In their 1996 study, Schiaffino et al. claimed that neural and hormonal factors could affect muscle development, leading to myogenic cell origin differences and the generation of primary/secondary fibers [119]. Slow muscle differentiation is broadly induced by neural influences from the spinal cord [120]. The contractile speed of re-innervated muscle, which can be fast or slow, may be affected by the different activities of distinct nerves [121].

Numerous clinical situations can cause denervation, including injury, aging, and cancer cachexia. The most frequent and serious outcomes of these disorders are muscle tissue malfunction and atrophy, and NMJ degradation could be a crucial factor in those processes [122]. Aging has a substantial effect on the neuromuscular system, reducing muscle function and motor ability. As individuals age, the NMJ deteriorates, resulting in poor signal transmission and muscular weakening [123,124]. In aged mice, DOK7 gene therapy enhances NMJ innervation, thereby enhancing muscles and motor activities [125]. Muscle stem cells (MuSCs) play an important role in muscle repair and regeneration. With age, these stem cells become less effective, contributing to muscular atrophy and decreased regeneration ability [126,127]. Aging MuSCs could be regenerated by potential treatments such as gene therapy or exercise targeting NMJ innervation [123,128].

### 5.3. Muscle to Nerve

The ways in which myokines influence the nervous system have been explored in several studies. Brain-derived neurotrophic factor (BDNF), synthesized primarily by the brain but also by skeletal muscle, supports muscle regeneration, β-oxidation in muscle tissue, insulin-regulated utilization of glucose, and NMJs [129]. BDNF can upregulate the proliferation of hippocampal neurons, neuronal plasticity, and synaptogenesis and inhibit neuroinflammation [130,131,132]. Irisin can regulate astrocytes and consequently provide neuroprotective effects for cultured neurons [133]. Moreover, Hashemi claimed that the neural differentiation rate of mouse embryonic stem cells is dramatically downregulated by FNDC5 (irisin) elimination [134].

Various mechanisms contribute to NMJ repair and maintenance. The formation, differentiation, and maintenance of NMJ regeneration require myofiber components and derived factors or associated satellite cells [135,136,137,138]. The depletion of skeletal stem cells could be a mechanism for NMJ decline [139].

**Table 2 ijms-26-07060-t002:** Muscle and nerve communication.

Factors	Effects on Nerve	Mechanism/Signaling Involved	Effects on Skeletal Muscle	Mechanism/Signaling Involved
Motor neurons			Clinical: Affecting muscle fiber morphology and phenotype [116,117] In vivo: Differentiation of slow muscles [120]; affects contractile speed of re-innervated muscle [121]	
Gap junctions/NMJs			In vitro: ↑ Myoblast fusion [118]	Intercellular communication [118]
			Poor signal transmission and muscular weakening in aging [123,124]	NMJ deteriorates, mitochondria mechanism [123,124]
Neural and hormonal influences			↑ Muscle development [119]	Isogenes [119]
DOK7			In vivo: ↑ Muscles and motor activities [125]	↑ NMJ innervation [125]
MuSCs	NMJ repair and maintenance [135,136,137,138,139]	Myofiber components, derived factors, associated satellite cells [135,136,137,138,139]	Muscle repair and regeneration [123,128]	
BDNF	↑ Hippocampal neurons, neuronal plasticity, and synaptogenesis ↓ Neuroinflammation [130,131,132]		Supporting muscle regeneration and utilization [129]	
Irisin	In vitro: Regulating astrocytes, neuroprotective effects [133]	Interleukins, COX-2, AKT, NF*κ*B [133]	In vitro: Mitochondrial biogenesis [82] In vivo: ↑ Myogenesis [83] ↓ Muscle atrophy [84]	
	In vitro: Neural generation and development [134]	Post-neural progenitor formation [134]		

NMJs, neuromuscular junctions; MuSCs, muscle stem cells; BDNF, brain-derived neurotrophic factor.

## 6. The Effects of PTH on Skeletal Muscle

Early English-language research papers on the internet focusing on PTH’s influence on skeletal muscle appeared in the early to mid-1980s [140,141]. However, it was not until the start of the 21st century that they became a trend, providing various in vitro, in vivo, and clinical insights. A detailed summary of laboratory experiments is provided in Table 3.

In vitro/ex vivo studies have used diverse muscle cell lines (C2, C2C12, and G8), rat skeletal muscle, and human skeletal muscle biopsies. Several experiments have established muscle cell lines composed of different types, such as C2C12 and fibro-adipogenic progenitor (FAP) cocultures, osteoblastic MC3T3-E1, and embryonic stem ZHTc6-MyoD. The doses of PTH (1–34) or (1–84) treatments covered a large range, from 10−12 to 10−6 mol/L, or from 0.1 pM up to 1000 nM. Generally, PTH can trigger skeletal muscle cell proliferation, migration, differentiation, and metabolism. Other PTH actions that have been reported are the browning of FAPs, the secretion of vascular endothelial growth factor, and Wnt signal inhibition [9,140,142,143,144,145,146].

Mice and rats are the two main models in all in vivo research studies. PTH (1–34) or PTH (1–84) is typically administered intermittently (3 times per week); the highest reported dose is 150 μg/kg body weight (BW)/day, and the lowest dose is 30 μg/kg BW. The therapeutic durations vary from a few days to several weeks. In general, the implications of PTH for skeletal muscle are obvious in pathological models: Mdx mice, OVX mice, and rats with rotator cuff tears [9,140,141,142,144,147,148,149,150].

In a prospective human cohort study of people aged 55 to 85 years, Visser found that the risk of sarcopenia was increased in adults with higher PTH levels and lower 25-hydroxyvitamin D levels [151]. In 2014, Sikjaer studied 62 chronic hypoparathyroidism patients treated with 100 μg of PTH (1–84) for 6 months and concluded that the treatment did not enhance quality of life and slightly diminished muscle strength [152]. Gielen investigated 518 European men aged 40–79 years and found that their gait speed, grip strength, and loss of muscle mass were not predicted by 25-hydroxyvitamin D or PTH levels; however, gait speed in men 70 years and older might be associated with low IGF-1 [153]. Palermo’s 2019 research compared 26 postmenopausal women with primary hyperparathyroidism with a group of 31 age- and body mass index-matched controls and reported a significant decrease in irisin concentration in postmenopausal women with primary hyperparathyroidism [145]. A cross-sectional study in Australia found major risk factors for sarcopenia, such as low phosphate and serum albumin, in 39 dialysis patients with a median age of 69 years; patients with sarcopenia showed slightly greater calcium, PTH, and 25OH-D levels than the non-sarcopenic group, but the links between those measures and sarcopenia were not statistically significant [154]. Overall, those clinical studies suggest that circulating PTH levels and sarcopenic skeletal muscle characteristics are not significantly associated.

## 7. Parathyroid Hormone, Bone, Nerves, and Skeletal Muscle

Bone and skeletal muscle exhibit a close interdependence via osteokines, myokines, and physical forces. Furthermore, evidence indicates that nerves and skeletal muscles have connections that affect each other in ways important to their mechanisms of action. Clinical neuromuscular diseases—conditions that cause skeletal muscle, motor nerve, or NMJ damage—are reported to raise the risk of bone fragility and fractures. Furthermore, nutritional problems, low physical activity or immobility, and the side effects of specific disease-modifying medication treatments in those patients negatively affect bone health [155]. One of the most prevalent endocrine pathologies is primary hyperparathyroidism, a condition caused by the excessive synthesis of PTH. Notably, this disorder is associated with bone mineral density reduction [156], impairs muscle strength [157], and leads to peripheral neurological abnormalities [158].

### 7.1. PTH’s Role in the Bone–Muscle Axis

PTH, FGF23, and 1,25(OH)2 vitamin D (1,25D) are fundamental to calcium and phosphate homeostasis and together create acknowledged endocrinologic feedback loops [159,160]. PTH is also reported to upregulate FGF23 expression through the PKA and Wnt pathways [161].

Some authors have recognized that PTH stimulates PGE2 production [162,163,164]. Especially when cortisol levels are low, the PGE2 released as a result of PTH effects is robust [163].

Several attempts have been made to clarify a link between PTH and osteocalcin. In 1997, Yu reported that the activation of the osteocalcin promoter caused by PTH was rapid and regulated via the cAMP-dependent protein kinase A pathway [165]. Three years later, Boguslawski et al. extended these research methods to determine other pathways. That team claimed that both PKA-dependent and PKC-dependent mechanisms mediate the regulation of osteocalcin transcription [166]. PTH significantly raises osteocalcin mRNA levels in MC3T3-E1 pre-osteoblast cells and the primary cultures of bone marrow stromal cells via multiple signaling pathways that require OSE1 and associated nuclear proteins [167].

IGF-1 plays a crucial role in mediating PTH’s anabolic effects on bone. As shown in both in vitro and in vivo experiments, IGF-1 mRNA and protein levels are upregulated by PTH [168,169,170]. Antibodies to IGF-1 prevent PTH and PTHrP from stimulating the formation of aggrecans in chondrocytes [171], and in in vitro experiments, PTH therapy has enhanced the number of cells in osteoblasts produced by IGF-1 knockout mice [172]. In the presence of IGF1R, PTH can act on bone formation [173] and activate osteoprogenitor cell proliferation and differentiation in mature osteoblasts [174]. Several signaling pathways involving IGF-1 and PTH have been shown to act in bone, including RANKL [174], ephrin B2/EphB4 [175], and PTH/Indian hedgehog [176].

The SOST gene encodes sclerostin and is inhibited by the intermittent or continuous administration of PTH in vitro and in vivo. Possible pathways and factors involved include the cAMP/PKA pathway and the proteasomal degradation of Runx2 [177,178]. After menopause, intermittent PTH treatment significantly downregulates serum sclerostin levels [179]. These data demonstrate that PTH stimulation decreases the production and secretion of sclerostin.

Data from multiple sources have identified a connection between PTH and IL-6. For example, PTH modulation was found to be responsible for the production of IL-6 in vitro and in vivo. Clinically, IL-6 serum levels are increased in patients with primary hyperparathyroidism [180,181,182].

Few studies have analyzed the relationship between PTH and myostatin. In fact, we could not find any in vitro or in vivo evidence concerning this relationship. In the STRAMBO study, Szulc found that the PTH serum level was not associated with myostatin concentrations [183]. Martino reported an indirect comparison in which the circulating levels of PTH and myostatin were negatively associated with the maximum voluntary contraction [184]. Further research is needed to clarify the connection between these factors.

Irisin is a novel myokine and adipokine that has attracted great attention lately. Studies have found that several metabolic actions of PTH are apparently opposed to those of irisin [185,186]. Palermo’s in vitro preclinical study reported the only findings about direct biological crosstalk between PTH and irisin [145]. Therefore, a new approach is needed to clarify the interactions between these two hormones.

Several studies have examined how PTH works with FGF2. In osteoporotic patients treated with PTH, serum FGF2 increases, and in osteoblasts, PTH elevates FGF2 mRNA and protein expression [187,188,189]. Additionally, although the osteoclastogenic effects of PTH are diminished in FGF2-null mice, the osteoclast-activating effect of PTH re-emerges in cells cocultured with osteoblasts or treated with FGF2. Therefore, PTH appears to increase osteoclast formation and bone resorption in mice, partly through endogenous FGF2 synthesis by osteoblasts [190].

Until recently, no reliable evidence has demonstrated any interactions between PTH and osteoactivin, IL-7/IL-15, or musclin.

### 7.2. PTH’s Role in the Nerve–Muscle Axis

Several biological agents and drugs exert dual effects on both skeletal muscle and nerve tissues, making them promising candidates for treating neuromuscular disorders. For instance, IGF-1 enhances muscle in both anabolic and catabolic pathways [48] and plays a significant role in nerve health and regeneration [191]. BDNF and GDNF (glial cell line-derived neurotrophic factor) are neurotrophic factors that stabilize neuromuscular junctions, promote neural health, and target skeletal muscle [192,193]. Testosterone and β2-adrenergic agonists contribute to muscle hypertrophy and nerve activities [194,195,196,197]. Additionally, exercise-induced myokines such as IL-6 and irisin mediate communication between muscles and nerves, influencing both regeneration and synaptic maintenance [198,199,200,201]. These molecules highlight the interconnected nature of muscle and nerve systems and are being explored for therapies targeting aging, sarcopenia, injury, and neurodegenerative diseases.

Calcium plays a crucial role in the function of NMJs: it is essential for neurotransmitter release, regulates muscle contraction, and maintains neuromuscular function [202,203,204]. Without Ca^2+^, ACh is not released and muscles do not contract. Systemic calcium imbalances, such as hypocalcemia or hypercalcemia, can thus negatively affect NMJs [205,206,207]. These clinical conditions might also be associated with PTH disorders (hypoparathyroidism or hyperparathyroidism). Abnormal MuSK expression, AChR clustering, and nerve branching can be caused by Ca^2+^ signal loss [204]. Lack of Ca^2+^ can also cause variations in NMJ structures and functions and induce a sustained stress response in muscles [208].

An in vitro study showed that PTH could boost the mean speed of both anterograde and retrograde organelle traffic on axons [209]. PTH (1–34) treatment can affect axonal regeneration by enhancing endogenous BMP-7 in rat Schwann cells [210]. Moreover, neurons in the subfornical organ—which lies above the third ventricle and modulates body fluid homeostasis—can be activated by circulating PTH [211].

ACh is a neurotransmitter that performs numerous roles in the brain and other organ systems and is commonly associated with the NMJ. Some authors have suggested a relationship between PTH and ACh. In the rat superior cervical ganglion (evaluated in vitro), ACh is released when PTH increases and calcitonin is inhibited [212]. The effect of PTH on ^3^H-acetylcholine synthesis in rat parathyroid glands has been investigated in vitro, and it was found to inhibit cholinergic activity [213]. Furthermore, ACh can be preserved by PTH-induced oxidative stress [214]. The effects of PTH on NMJ components are summarized in Table 4.

As noted, agrin, rapsyn, LRP4, and MuSK are important parts of the NMJ. LRP4 can also be involved in Wnt signaling activity, which can influence skeletal muscle. Nevertheless, little research has directly linked PTH with these NMJ components.

**Table 4 ijms-26-07060-t004:** PTH’s actions on NMJ components.

Target of NMJ Components	Study Design	PTH’s Effects	Ref.
Axon/Neuron	In vitro	PTH boosts the mean speed of both anterograde and retrograde organelle traffic on axons	[209]
	In vivo	PTH (1–34) treatment can affect axonal regeneration by enhancing endogenous BMP-7 in rat Schwann cells	[210]
	In vivo	Circulating PTH activates neurons in the subfornical organ	[211]
Acetylcholine activities	In vitro	In the rat superior cervical ganglion, ACh is released when PTH increases and calcitonin is inhibited	[212]
	In vitro	PTH affects 3H-acetylcholine synthesis in rat parathyroid glands	[213]
	In vitro	PTH-induced oxidative stress preserves ACh	[214]

PTH, parathyroid hormone; BMP, bone morphogenetic protein; ACh, acetylcholine.

### 7.3. PTH and Skeletal Muscle

Skeletal muscle metabolism is influenced by both systemic and local regulators that ensure proper function, growth, and repair. As a systemic endocrine hormone, PTH not only affects muscle cells directly, but also induces local factors that influence muscle metabolism indirectly.

PTH has significant effects on skeletal muscle, influencing both muscle function and metabolism. In vitro evidence shows that PTH affects muscle cell proliferation, migration, and differentiation. PTH modulates the release of alanine and glutamine and the muscle cell uptake and retention of 25(OH)D3. PTH activities in vivo include changes in muscle mass and strength, muscle fiber type transitioning, reductions in inflammatory gene expression, and muscle atrophy inhibition. Most of the preclinical studies have not verified the direct action of PTH on muscle via its receptors. Further research is needed to determine the specific effects of PTH in terms of the duration of action and effects of concentration. In clinical settings, nearly all published research has focused on comparing changes in PTH serum levels with skeletal muscle features; however, the results seem to indicate no significant connection. PTH is known for its instability; it degrades relatively quickly, requiring the careful handling of blood samples to ensure accurate measurements. Proper pre-analytical conditions, including specimen type, sampling time, and storage, are crucial for reliable PTH testing [215]. Furthermore, PTH has immunomodulatory effects, impacting various immune cells and functions, but the clinical significance remains unclear [216]. While PTH replacement medication, such as teriparatide, is usually considered safe and effective for treating hypoparathyroidism, it can have side effects (most notably, hypercalcemia) [217]. When used for fracture healing, PTH has a well-established safety profile, with mild side effects such as bruising at the injection site [218]. Evidence is still lacking on how PTH treatment can transform skeletal muscle characteristics, which is surprising given that several PTH products have already been FDA-approved. This leaves a gap for further investigations on the effects of PTH and its detailed mechanisms in humans.

Recent studies have revealed that extracellular vesicles (EVs)—particularly exosomes—and their cargo of extracellular RNAs (exRNAs) are central players in the communication between skeletal muscle, bone, and the nervous system. These vesicles shuttle bioactive molecules—including regulatory microRNAs—that influence processes such as osteogenesis, synaptic maintenance, and myogenesis. Growing evidence suggests that muscle- and bone-derived EVs can cross-regulate each other’s tissues, shaping local regeneration and systemic homeostasis [219,220,221]. Importantly, PTH has emerged as a modulator of EV composition, potentially altering EV-derived miRNA profiles to influence bone turnover [143,222,223]. This evolving area offers a novel perspective beyond traditional hormone signaling, pointing to PTH-EV-miRNA axes as promising therapeutic targets in musculoskeletal and neuromuscular disorders.

Chronic PTH elevation—as observed in primary or secondary hyperparathyroidism—leads to bone complications, muscle atrophy and weakness, and neuromuscular manifestations [184,224,225]. This persistent exposure status may disrupt calcium and phosphorus balance [184] and trigger endothelial dysfunction [214]. In contrast, intermittent administration of PTH, such as daily or every-other-day injections of teriparatide (PTH 1–34), has shown anabolic effects on the skeleton. This mode of delivery mimics the physiological pulsatility of PTH and is associated with increased bone functional outcomes in both animal models and clinical observations [185]. Likewise, many studies have shown the benefits of this method on the skeletal muscle, as we have summarized and discussed in the previous sections. These bases underscore the critical importance of exposure patterns; while chronic exposure is detrimental to musculoskeletal and neural health, intermittent PTH may be protective and pro-regenerative, supporting careful therapeutic titration in musculoskeletal disease contexts.

Recent advances suggest that organ-to-organ communication mediated by PTH signaling can be leveraged for therapeutic purposes. PTH acts as a central regulator at the interface of bone, muscles, and nerves, engaging shared molecular signaling axes that coordinate tissue function and regeneration. The involvement of these three tissues in key regulators such as Wnt [226,227], IGF-1/AKT [47,48,191], MAPK [228,229,230], and the calcineurin-NFAT pathway [231,232] highlights a multi-organ crosstalk mechanism. Targeting these shared pathways offers the potential to simultaneously modulate osteogenesis, myogenesis, and neuromuscular synaptic maintenance. Bone-, muscle-, and nerve-derived factors like IGF-1, irisin, and BDNF act systemically to promote musculoskeletal health and regeneration and support neuromuscular junction structure. Therapeutic strategies aimed at fine-tuning PTH delivery (e.g., pulsatile analogs) or modulating PTH-induced content may open new avenues for treating age-related musculoskeletal and NMJ degeneration, particularly in conditions like osteosarcopenia or sarcopenic neurodegeneration. The complex relationships between PTH, muscle, bone, and nerves are the subjects of ongoing investigations and have implications for aging, metabolic disorders, and musculoskeletal diseases.

## 8. Conclusions and Future Directions

As a systemic endocrine regulator, PTH plays a crucial role in the physiology of skeletal muscle, both directly and through interactions with bone and the nervous system via modulating osteokines, myokines, and neuromuscular activity. The comprehensive crosstalk between bones and muscles and between nerves and muscles highlights the importance of hormonal balance for musculoskeletal health. Understanding the precise molecular mechanisms through which PTH affects skeletal muscle could lead to new therapeutic approaches for preventing and treating skeletal muscle-related disorders, particularly in conditions such as hyperparathyroidism, chronic kidney disease, and age-related muscle loss.

In this review, we highlight the potential to investigate the effects of PTH on osteokines, myokines, neuromuscular junctions, and other new modulators (e.g., EV-derived miRNAs) that may positively influence the musculoskeletal and neuromuscular axes. Clinical studies focused directly on skeletal muscle, which were previously limited, can now be designed to examine changes in the relevant organs. Future research should continue to explore how PTH-targeted therapies can optimize muscle function while minimizing potential adverse effects. By addressing these complex interactions, novel strategies can be developed to improve mobility, strength, and overall quality of life in affected individuals.

## Figures and Tables

**Figure 1 ijms-26-07060-f001:**
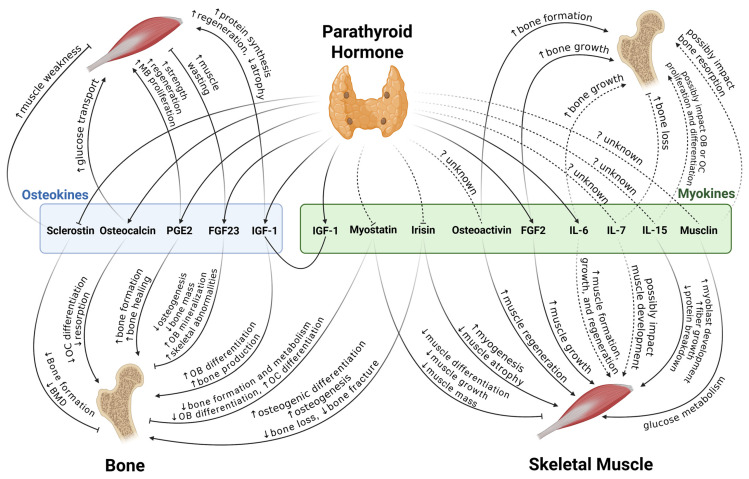
PTH acts on osteokines and myokines. Arrow: positive effects; inhibitor: negative effects; solid line: strong evidence and consensus; dashed line: controversial or unclear. PGE2, prostaglandin E2; FGF, fibroblast growth factor; IGF, insulin-like growth factor; IL, interleukin; MB, myoblast; BMD, bone mineral density; OC, osteoclast; OB, osteoblast.

**Figure 2 ijms-26-07060-f002:**
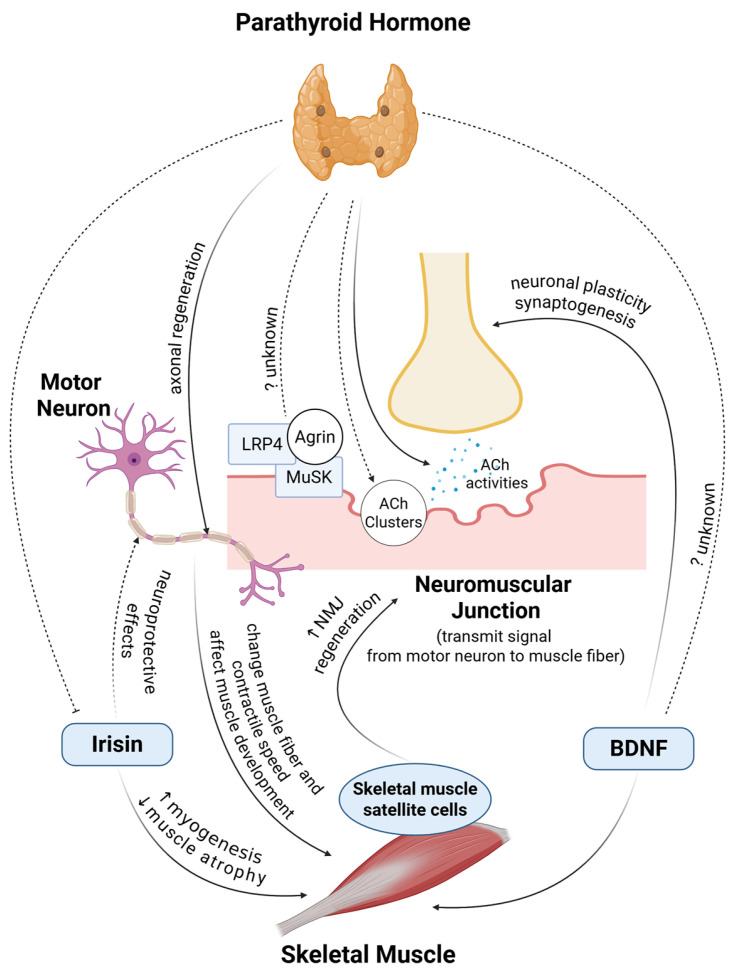
High Potential for PTH to enhance the NMJ and its components. Arrow: positive effects; inhibitor: negative effects; solid line: strong evidence and consensus; dashed line: controversial or unclear. LRP, low-density lipoprotein receptor-related protein; MuSK, muscle-specific kinase; NMJ, neuromuscular junction; ACh, acetylcholine; BDNF, brain-derived neurotrophic factor.

**Table 3 ijms-26-07060-t003:** A compilation summarizing the effects of PTH on skeletal muscle.

**In Vitro/Ex Vivo**			
**Treatment**	**Cell Model**	**Mechanism**	**Ref.**
100 nM PTH	FAPs and C2C12 coculture	- Promotes browning of FAPs and inhibits the accumulation of lipid droplets - PTH-induced browning of FAPs promotes myogenic differentiation by secretion of VEGF	[142]
PTH (1–84) from 10^−6^ to 10^−12^ mol/L	hSCs	Promotes the myogenic differentiation process	[143]
1–1000 nM PTH (1–34)	3T3-L1, MCF7, C2C12, MC3T3-E1, G8	- Induces cell proliferation, migration, and differentiation while reducing lipid secretion in myoblasts - Wnt signal inhibition blocks PTH-induced cell proliferation and migration and reduces lipid secretion in myoblasts	[144]
10^−10^ M to 10^−8^ M PTH (1–34)	C2C12, MC3T3-E1	r-Irisin leads to a 50% downregulation of PTH-r mRNA expression compared with untreated cells	[145]
0.1 pM, 1 pM, 10 pM and 100 pM PTH (1–34)	C2	Modulates muscle cell uptake and retention of 25(OH)D3	[146]
20 nM rat PTH (1–34)	C2C12 and ZHTc6-MyoD	Accelerates myocyte differentiation	[9]
PTH (1–34) and PTH (1–84)	Rat skeletal muscle	Increases the release of alanine and glutamine	[140]
**In vivo**			
**Treatment**	**Animal model**	**Mechanism**	**Ref.**
30 μg/kg teriparatide 3 times a week for 4 or 8 weeks	9-week-old Sprague Dawley rats	- Inhibits muscle atrophy and fatty infiltration - Promotes the expression of UCP1 (after rotator cuff tears)	[142]
80 μg/kg of PTH (1–34) three times a week for 20 weeks	8-week-old female WT C57BL/6J mice	- Abrogates OVX-induced reduction in exercise performance and lactate metabolism - Improves oxidative fiber ratio, cross-sectional area, and intramyocellular lipids	[144]
30 µg/kg TPTD, 3 days/week	7-month-old female Wistar rats	Improves bone, skeletal muscle, and fat mass	[147]
150 μg/kg body weight/day of PTH (1–34), daily	4-week-old C57BL/10ScSn-Dmd^mdx^/J (Mdx) and C57BL/10SnJ wild-type (WT), male mice	- Decreases inflammatory gene expression in skeletal muscle - Increases type 1 and decreases type 2c skeletal muscle fibers in Mdx mice	[148]
PTH (1–34) 60 μg/kg/day, 5 days a week	12–14-week-old female Wistar rats	- PTH alone had no effect on muscle mass - PTH + GH was more efficient at preventing loss of bone strength, density, and micro-architecture	[149]
60 μg/kg/d of PTH for 59 days	4-week-old Mdx mice	Improved the muscle strength and histological characteristics of the skeletal muscle	[9]
PTH (40 μg/kg BW/day) every other day for 1–35 days	3-month-old female Sprague–Dawley rats	No impacts on muscle weight or muscle fiber size	[150]
1–84 or 1–34 PTH, 200 U/day, for 4 days	Sprague Dawley rats weighing 150 to 200 g	Decreases energy production, transfer, and utilization	[141]
PTH (1–34) and PTH (1–84)	Sprague Dawley rats	In primary hyperparathyroidism and chronic uremia, PTH may directly impact muscle dysfunction and wasting	[140]

PTH, parathyroid hormone; FAPs, fibro-adipogenic progenitors; VEGF, vascular endothelial growth factor; hSCs, human skeletal muscle biopsies; mRNA, messenger RNA; UCP1, uncoupling protein 1; WT, wild type; OVX, ovariectomy; TPTD, teriparatide; GH, growth hormone; BW, body weight.

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
