# Peer review of "Parathyroid Hormone as a Modulator of Skeletal Muscle: Insights into Bone–Muscle and Nerve–Muscle Interactions"

_ijms, 2025, doi:10.3390/ijms26157060_

Round 1

Reviewer 1 Report (Previous Reviewer 2)

Comments and Suggestions for Authors

Parathyroid Hormone as a Modulator of Skeletal Muscle: Insights into Bone–Muscle and Nerve–Muscle Interactions

2nd revision

Thank you for addressing the comments raised in the previous review and for submitting your revised manuscript. I have carefully considered your responses and the changes made.

While I appreciate the efforts made, I find that some of the major concerns previously highlighted have not yet been fully resolved to a satisfactory extent. Please see my detailed feedback below:

  1. Clarity of Objective and Novel Contribution
  • My previous comment: "The manuscript's specific objective and its intended novel contribution to the existing literature are not immediately clear. Could the authors explicitly state the primary goal of this review and highlight what new perspective or synthesis it offers beyond previous work? A clearer focus would help frame the presented information more effectively."
  • Authors' Response: "At the end of the introduction, we present the relationship between PTH, muscle, bone and nerve which may open up new steps in the selection of treatment methods for related diseases. We think that our current paper is unique and opens up some specific research directions for PTH. (Line 61–65). Given the intricate relationships among PTH, bone, nerve, and muscle, understanding the systemic and local effects of PTH on skeletal muscle is crucial for developing targeted therapeutic strategies for various conditions. Investigating the molecular pathways involved in PTH-mediated muscle regulation could provide valuable insights into musculoskeletal disorders, particularly those associated with aging, metabolic disease, and endocrine dysfunction."
  • My current feedback: Thank you for pointing to lines 61-65. While this section was present in the original manuscript and outlines the general importance of the topic, it still does not explicitly articulate the specific novel contribution of this review. What unique synthesis, perspective, or gap in the existing literature does this review address that has not been covered by previous reviews? A more direct statement of the manuscript's unique value proposition would greatly enhance its impact.
  1. Methodology for Literature Search
  • My previous comment: "As this is presented as a review article, transparency regarding the literature search and selection process is crucial. The review would be significantly strengthened by including details about the methodology used (e.g., databases searched, search terms and combinations, inclusion/exclusion criteria). This allows readers to understand the scope of the review and assess the potential risk of overlooking relevant studies."
  • Authors' Response: "We added a Methodology part. (Line 66–72) Investigations of publications and reviews were performed on the PubMed database to discover relevant studies published until 2024. The database was queried using the following keywords: parathyroid hormone, PTH, skeletal muscle, bone–muscle, nerve–muscle, myokines, osteokines, neuromuscular junction (NMJ), and the combinations of these keywords. Besides, we listed detailed keywords of the names of myokines and osteokines and combined them with the mentioned keywords to support deep searching in the fields we are interested in. Additionally, pertinent references cited in the previously discovered works were examined."
  • My current feedback: I acknowledge the addition of the 'Methodology' section (Lines 66-72). While this is a step in the right direction, the description provided is somewhat brief. Furthermore, relying solely on PubMed for a comprehensive review may introduce a selection bias, potentially omitting relevant studies indexed in other databases (e.g., Scopus, Web of Science, Embase). Expanding the search strategy to include other major databases and providing more detail on the inclusion/exclusion criteria would strengthen the methodological rigor of the review.

Regarding the other major points raised previously—Depth and Organization of Content, and Timeliness of References—it appears these were not addressed with significant manuscript revisions. I will reiterate them here with comments on your responses:

  1. Depth and Organization of Content
  • My previous comment: "Several sections seem somewhat underdeveloped or presented in a way that could be confusing for the reader. There appears to be a mix of information from different study types (animal models, cell studies, human data) and contexts (e.g., exercise, various pathologies) without always clearly distinguishing between them or providing sufficient depth. A more thorough exploration of the topics and a potential reorganization of the material could enhance clarity and reader engagement. For instance, distinguishing clearly between in vitro/animal findings and human physiological/pathological data is essential throughout the text."
  • Authors' Response: "In this review, we present the interaction of PTH directly on muscle and indirectly through the musculoskeletal and musculo-neural connections. The content of musculoskeletal and neuromuscular interactions is summarized as concisely and concisely as possible, so that many different studies on in vitro, in vivo or clinical studies can be presented in a non-unified manner in the sub-sections. One of the reasons for this is that we could not find a complete system with appropriate studies to present in a unified and clear manner. We try to supplement the reviewer's comments below in the most specific and clear way."
  • My current feedback: I understand your rationale that providing a concise summary of diverse study types was the aim. However, the lack of clear distinction between findings from different models (in vitro, animal, human) and contexts (e.g., exercise, pathology) can still lead to confusion for the reader. A more structured approach, perhaps with dedicated sub-sections or clearer signposting within the text to differentiate these evidence types, would significantly improve readability and the critical appraisal of the evidence presented.
  1. Timeliness of References
  • My previous comment: "While the references cited are generally relevant, a notable portion (approximately 60%) are more than ten years old. Although citing foundational studies is often necessary, the review would benefit from incorporating more recent findings to reflect the latest advancements in this rapidly evolving field. An updated literature search could enhance the manuscript's currency and impact."
  • Authors' Response: "We have tried our best to update the references. An important part of our report is neuromuscular interaction, which seems to have been studied for a long time and has many consensus conclusions in the literature. Besides, many new studies have produced quite similar results, because of the volume of the report, we focused on selecting representative studies with high reliability. These lead to our limitations."
  • My current feedback: I acknowledge your explanation regarding the foundational nature of some older studies, particularly concerning neuromuscular interaction. However, the field is dynamic. While foundational studies are important, a review article also benefits greatly from integrating the very latest research (within the last 5-10 years) to ensure it provides a current and comprehensive overview. A more concerted effort to incorporate recent high-impact publications would enhance the manuscript's relevance and demonstrate a thorough engagement with contemporary literature.

In summary, while some progress has been made, the manuscript still requires substantial revisions to address these fundamental points before it can be considered suitable for publication. I encourage you to undertake these revisions, as the topic is of interest and a well-structured, comprehensive review would be a valuable contribution.

Author Response

2nd revision

Thank you for addressing the comments raised in the previous review and for submitting your revised manuscript. I have carefully considered your responses and the changes made.

While I appreciate the efforts made, I find that some of the major concerns previously highlighted have not yet been fully resolved to a satisfactory extent. Please see my detailed feedback below:

Clarity of Objective and Novel Contribution

My previous comment: "The manuscript's specific objective and its intended novel contribution to the existing literature are not immediately clear. Could the authors explicitly state the primary goal of this review and highlight what new perspective or synthesis it offers beyond previous work? A clearer focus would help frame the presented information more effectively."

Authors' Response: "At the end of the introduction, we present the relationship between PTH, muscle, bone and nerve which may open up new steps in the selection of treatment methods for related diseases. We think that our current paper is unique and opens up some specific research directions for PTH. (Line 61–65). Given the intricate relationships among PTH, bone, nerve, and muscle, understanding the systemic and local effects of PTH on skeletal muscle is crucial for developing targeted therapeutic strategies for various conditions. Investigating the molecular pathways involved in PTH-mediated muscle regulation could provide valuable insights into musculoskeletal disorders, particularly those associated with aging, metabolic disease, and endocrine dysfunction."

My current feedback: Thank you for pointing to lines 61-65. While this section was present in the original manuscript and outlines the general importance of the topic, it still does not explicitly articulate the specific novel contribution of this review. What unique synthesis, perspective, or gap in the existing literature does this review address that has not been covered by previous reviews? A more direct statement of the manuscript's unique value proposition would greatly enhance its impact.

- We fixed our introduction following your feedback. We hope the updated one could better emphasize our aim in the review article (1. Introduction, Line 28-45).

  1. Introduction

Parathyroid hormone (PTH) is a critical endocrine regulator of calcium and phosphate homeostasis, primarily known for its effects on bone and kidney metabolism [1]. Recent studies have highlighted its broader role in musculoskeletal health, particularly its effects on skeletal muscle. Meanwhile, bone and muscle are closely interconnected through both mechanical loading and biochemical signaling, playing key roles in muscle metabolism and function [2,3]. Furthermore, the nervous system is essential for maintaining muscle mass and function as neuromuscular signaling regulates muscle contraction, protein synthesis, and energy metabolism through calcium control [4,5]. Considering the classical role of PTH and the muscle symptoms (muscle weakness, myopathy, etc.) that occur in hyperparathyroidism and hypoparathyroidism, it is thought that PTH may affect these musculoskeletal and neuromuscular interactions in a complex manner [6]; however, there is a lack of comprehensive information on the systemic and local effects of PTH on skeletal muscle. Investigating the molecular pathways involved in PTH-mediated muscle regulation is expected to provide valuable insights into musculoskeletal diseases—particularly those related to aging, metabolic disorders, and endocrine dysfunction—and to suggest targeted treatment strategies for various musculoskeletal conditions.

Our review synthesizes emerging evidence on the non-classical roles of PTH in skeletal muscle and neuromuscular junction function. By integrating findings from clinical, in vivo, and in vitro studies, we provide a novel perspective on PTH as a potential therapeutic modulator in the muscle–bone–nerve crosstalk, addressing a critical gap in current musculoskeletal research.

Methodology for Literature Search

My previous comment: "As this is presented as a review article, transparency regarding the literature search and selection process is crucial. The review would be significantly strengthened by including details about the methodology used (e.g., databases searched, search terms and combinations, inclusion/exclusion criteria). This allows readers to understand the scope of the review and assess the potential risk of overlooking relevant studies."

Authors' Response: "We added a Methodology part. (Line 66–72) Investigations of publications and reviews were performed on the PubMed database to discover relevant studies published until 2024. The database was queried using the following keywords: parathyroid hormone, PTH, skeletal muscle, bone–muscle, nerve–muscle, myokines, osteokines, neuromuscular junction (NMJ), and the combinations of these keywords. Besides, we listed detailed keywords of the names of myokines and osteokines and combined them with the mentioned keywords to support deep searching in the fields we are interested in. Additionally, pertinent references cited in the previously discovered works were examined."

My current feedback: I acknowledge the addition of the 'Methodology' section (Lines 66-72). While this is a step in the right direction, the description provided is somewhat brief. Furthermore, relying solely on PubMed for a comprehensive review may introduce a selection bias, potentially omitting relevant studies indexed in other databases (e.g., Scopus, Web of Science, Embase). Expanding the search strategy to include other major databases and providing more detail on the inclusion/exclusion criteria would strengthen the methodological rigor of the review.

Regarding the other major points raised previously—Depth and Organization of Content, and Timeliness of References—it appears these were not addressed with significant manuscript revisions. I will reiterate them here with comments on your responses:

- We updated our search strategy and methodology for more details and varied our search in all the big databases (2. Methodology, Line 46-65).

  1. Methodology

A comprehensive literature search was conducted to identify relevant studies published up to 2025, with a focus on the role of parathyroid hormone (PTH) in skeletal muscle and its interactions with bone and nerve systems. The primary database used was PubMed, supplemented by Scopus, Web of Science, ScienceDirect, and Embase to ensure broad coverage of biomedical and interdisciplinary research.

The following keywords and their combinations were used in the search: parathyroid hormone, PTH, skeletal muscle, hormones, bone–muscle, nerve–muscle, myokines, osteokines, neuromuscular junction, and NMJ. In addition, we included the names of specific myokines and osteokines (e.g., IL-6, irisin, FGF23, and osteocalcin) and NMJ components (e.g., acetylcholine cluster, Agrin, and MuSK) to refine and deepen the search. Boolean operators (AND, OR) were applied to structure complex queries and optimize retrieval.

The inclusion criteria were as follows:

- Original research articles, reviews, and meta-analyses published in English.

- Studies addressing the role or effects of PTH (or its analogs) on skeletal muscle, neuromuscular junctions, or crosstalk between muscle and bone or nerve.

- Studies involving in vivo, in vitro, or clinical investigations.

The exclusion criteria were non-peer-reviewed sources, editorials, and conference abstracts without primary data.

To ensure the depth and accuracy of the review, reference lists of key publications were also manually examined to identify additional pertinent studies. All retrieved records were screened based on title and abstract, and a full-text review was conducted to confirm eligibility.

My previous comment: "Several sections seem somewhat underdeveloped or presented in a way that could be confusing for the reader. There appears to be a mix of information from different study types (animal models, cell studies, human data) and contexts (e.g., exercise, various pathologies) without always clearly distinguishing between them or providing sufficient depth. A more thorough exploration of the topics and a potential reorganization of the material could enhance clarity and reader engagement. For instance, distinguishing clearly between in vitro/animal findings and human physiological/pathological data is essential throughout the text."

Authors' Response: "In this review, we present the interaction of PTH directly on muscle and indirectly through the musculoskeletal and musculo-neural connections. The content of musculoskeletal and neuromuscular interactions is summarized as concisely and concisely as possible, so that many different studies on in vitro, in vivo or clinical studies can be presented in a non-unified manner in the sub-sections. One of the reasons for this is that we could not find a complete system with appropriate studies to present in a unified and clear manner. We try to supplement the reviewer's comments below in the most specific and clear way."

My current feedback: I understand your rationale that providing a concise summary of diverse study types was the aim. However, the lack of clear distinction between findings from different models (in vitro, animal, human) and contexts (e.g., exercise, pathology) can still lead to confusion for the reader. A more structured approach, perhaps with dedicated sub-sections or clearer signposting within the text to differentiate these evidence types, would significantly improve readability and the critical appraisal of the evidence presented.

- We have designed additional tables 1 and 2, which summarize the study designs and results and the pathways, signalings, and mechanisms involved. We hope that these two summary tables can clarify and provide a concise overview of the contents we present in the relevant sections (Table 1: Line 232-233, Table 2: Line 293-294).

Table 1. Bone and skeletal muscle interactions.

Osteokines/Myokines

Effects on Bone

Mechanism/Signaling Involved

Effects on Skeletal Muscle

Mechanism/Signaling Involved

FGF23

In vitro: ↓ BMSCs osteogenesis, ↓ mature osteoblast mineralization [27]

FGFR3-ERK

In vitro: ↓ Muscle cell differentiation [30]

Insulin/IGF-1, klotho

Clinical: Serum level associated with low bone mass [28]

Relevant to skeletal muscle wasting [33]

PGE2

Regulating both bone resorption and formation processes [34,35]

In vitro: Myogenesis [38]

EP4 receptor

In vivo, in vitro: Muscle regeneration and strength [39]

Muscle-specific stem cells

Related: Wnt, β-catenin [36], cAMP/PKA [37]

Osteocalcin

Glucose metabolism, reproduction, and cognition [41]

In vitro: ↑ Glucose transport [43]

Clinical, in vivo: ↑ Muscle uptake [44,45]

In vitro: ↓ Osteoclasts differentiation [42]

GPRC6A

IGF-1

↑ Osteoblasts differentiation, bone production [47]

↑ Protein synthesis and regeneration

↓ Muscle atrophy [101]

Related: PI3K/Akt, MAPK/ERK [46]

Sclerostin

Clinical, in vivo, in vitro: ↓ Bone formation [50,51]

Wnt [49]

Clinical: ↑ Muscle weakness [54]

Myostatin

↓ Bone formation

Metabolism [57]

↓ Muscle mass [55]

In vitro: ↓ Osteoblastic differentiation [58]

Osteocyte-derived exosomal miR-218

↓ Osteoclast differentiation [59]

RANKL, NFATC1

Related: ERK1/2, Wnt, TGF-β1, IGF-1 [55]

Osteoactivin

In vivo, in vitro: ↓ Osteoclastogenesis [102]

CD44-ERK

In vivo: Protection from fibrosis [63]

MMP-3, MMP-9

In vivo, in vitro: ↑ Bone formation [66]

TGF-β

IL-6

↑ Bone growth [70,71]

↑ Formation, growth, regeneration, satellite-cell-dependent myogenesis [69]

↑ Bone loss (in several osteolytic diseases) [70,71]

↑ Protein synthesis and breakdown

Engaged with muscle atrophy [69]

IL-7

In vivo: ↑ Bone loss [74]

RANKL

Might affect satellite cells [72,73]

IL-15

Bidirectional regulatory function [79]

Clinical: ↑ Myoblast development, fiber growth; ↓ protein breakdown [78]

Irisin

In vivo: ↓ Bone loss [84]

In vitro: Mitochondrial biogenesis [82]

In vivo:

↑ Myogenesis [83]

↓ Muscle atrophy [84]

Related: MAPK [85], ERK/STAT, BMP/SMAD [87], Wnt/β-catenin [88]

FGF2

Bone growth [92–94]

Clinical, in vivo: ↑ Muscle growth, intramuscular adipogenesis [91]

miR-29a/SPARC

In vivo: Bone marrow MSC Osteogenesis

ERK/Runx2 [95]

Musclin

Bone resorption [100]

RANKL

Glucose metabolism [96,97]

In vivo: ↑ Physical endurance [98]

Mitochondrial biogenesis

BMSCs, bone marrow mesenchymal stem cells; GPRC6A, G protein-coupled receptor class C group 6; RANKL, receptor activator of nuclear factor κB ligand; NFATC1, nuclear factor of activated T-cells; MMP, matrix metalloprotease; MSCs, mesenchymal stem cells.

Table 2. Muscle and nerve communication.

Factors

Effects on Nerve

Mechanism/Signaling Involved

Effects on Skeletal Muscle

Mechanism/Signaling Involved

Motor neurons

Clinical: Affecting muscle fiber morphology and phenotype [118,119]

In vivo: Differentiation of slow muscles [122]; affects contractile speed of re-innervated muscle [123]

Gap junctions / NMJs

In vitro: ↑ Myoblast fusion [120]

Intercellular communication

Poor signal transmission and muscular weakening in aging [125,126]

NMJ deteriorates, mitochondria mechanism

Neural and hormonal influences

↑ Muscle development [121]

Isogenes

DOK7

In vivo: ↑ Muscles and motor activities [127]

↑ NMJ innervation

MuSCs

NMJ repair and maintenance [137–141]

Myofiber components, derived factors,

associated satellite cells

Muscle repair and regeneration [125,130]

BDNF

↑ Hippocampal neurons, neuronal plasticity, and synaptogenesis

↓ Neuroinflammation [132–134]

Supporting muscle regeneration and utilization [131]

Irisin

In vitro: Regulating astrocytes, neuroprotective effects [135]

Interleukins, COX-2, AKT, NFκB

In vitro: Mitochondrial biogenesis [82]

In vivo:

↑ Myogenesis [83]

↓ Muscle atrophy [84]

In vitro: Neural generation and development [136]

Post-neural progenitor formation

NMJs, neuromuscular junctions; MuSCs, muscle stem cells; BDNF, brain-derived neurotrophic factor.

Timeliness of References

My previous comment: "While the references cited are generally relevant, a notable portion (approximately 60%) are more than ten years old. Although citing foundational studies is often necessary, the review would benefit from incorporating more recent findings to reflect the latest advancements in this rapidly evolving field. An updated literature search could enhance the manuscript's currency and impact."

Authors' Response: "We have tried our best to update the references. An important part of our report is neuromuscular interaction, which seems to have been studied for a long time and has many consensus conclusions in the literature. Besides, many new studies have produced quite similar results, because of the volume of the report, we focused on selecting representative studies with high reliability. These lead to our limitations."

My current feedback: I acknowledge your explanation regarding the foundational nature of some older studies, particularly concerning neuromuscular interaction. However, the field is dynamic. While foundational studies are important, a review article also benefits greatly from integrating the very latest research (within the last 5-10 years) to ensure it provides a current and comprehensive overview. A more concerted effort to incorporate recent high-impact publications would enhance the manuscript's relevance and demonstrate a thorough engagement with contemporary literature.

- We made every effort to keep the references up-to-date this time. Older references are mostly conceptual, historical, and unique research. We would like to present an overview of the proportions of all references here. We hope that this proportion is acceptable.

Period of Time

Quantity

%

2020-2025

66

28.2

2015-2019

62

26.5

2010-2014

38

16.2

2000-2009

38

16.2

Before 2000

30

12.8

Total

234

In summary, while some progress has been made, the manuscript still requires substantial revisions to address these fundamental points before it can be considered suitable for publication. I encourage you to undertake these revisions, as the topic is of interest and a well-structured, comprehensive review would be a valuable contribution.

We appreciate your comments and feedback. We strive to improve our reviews as much as possible and provide a constructive perspective for future research. We are grateful very much.

Reviewer 2 Report (New Reviewer)

Comments and Suggestions for Authors

This article addresses an important and timely topic - the bidirectional dialogue between the nervous and skeletal systems, with a particular focus on parathyroid hormone (PTH). While the review condenses a disparate collection of studies and introduces interesting molecular players, it is afflicted by conceptual focus, logical synthesis, as well as structure and linguistic clarity. It needs extensive revision before consideration for publication.

Major Comments

1. The paper currently reads more like an encyclopedic collection or textbook chapter than a hypothesis-driven scientific review. Much of what it contains is peripheral or general to the main subject and does not meaningfully contribute to a distinct argument. Authors are strongly urged to state the major goal or key hypothesis in early portions of the introduction, develop an unambiguous conceptual framework to govern the discussion, and to avoid background information except where it is directly pertinent to rationale or therapeutic implication of PTH for neuromuscular interaction.

2. The skeletal muscle biology, hormone regulation, and bone-muscle-nerve physiology sections are informative but highly descriptive. These can be significantly condensed and reorganized to provide only essential background information for appreciation of novelty and scope of the review.

3. The section on the molecular biology of PTH and localization to receptors is too lengthy and just recounts familiar facts without strategic importance to the major objectives of the review. The latter ought to be focused on the functional importance of PTH in relation to bone-muscle-nerve signaling, emphasize recent advances and unresolved mechanistic insight, and minimize well-documented endocrine activities unless directly relevant to muscle/nerve crosstalk.

4. Rather than enumerating facts within distinct subtopics, the authors can aim at conceptual integration. For example: compare the effects of intermittent versus chronic PTH exposure on neuromuscular health, integrate bone-derived and nerve-mediated mechanisms to display shared axes of signaling altered by PTH, include visual conceptual models (e.g., diagrams, summary tables) for targets for treatment, signal transduction pathways, or organ-to-organ communication.

5. The discussion of PTH-based therapies is overly optimistic. The authors need to balance the discussion by including translational challenges such as: Delivery methods (e.g., instability of peptides, targeting tissues), Immunogenicity, Safety and off-target effects over the long-term, Clinical trial design difficulties in neuromuscular diseases

6. The mention of EV-derived miRNAs and other new modulators is intriguing but underdeveloped. Mention of these features should be prominent, contextually linked to PTH-induced nerve or muscle effects, and supported by recent high-impact references.

7. The conclusion is typically a summary of earlier content. It should declare primary mechanistic shortcomings in PTH signaling in bone, nerve, and muscle, outline experimental or clinical approaches to fill them, and indicate areas of highest priority for future therapeutic research.

8. The figures are unreadable and require an increased image resolution, larger font size, better labeling of molecular pathways and their relationship with PTH signaling.

Additional Recommendations:

1. Consider about dividing the review into mechanistic segments (e.g., PTH's role in bone-muscle, nerve-muscle, and bone-nerve axes) for thematic consistency.

2. Insert a table summarizing comparison of experimental models (in vitro, in vivo, clinical) and their principal findings on PTH's neuromuscular actions.

3. Describe if muscle-specific or nerve-specific administration of PTH analogs has been examined.

Comments on the Quality of English Language

The paper is full of grammatical, stylistic, and structural issues that impede comprehension. Sentences are too often awkwardly phrased or unnecessarily convoluted. A professional copy editing by a native English-speaking scientific editor is strongly advised.

Author Response

This article addresses an important and timely topic - the bidirectional dialogue between the nervous and skeletal systems, with a particular focus on parathyroid hormone (PTH). While the review condenses a disparate collection of studies and introduces interesting molecular players, it is afflicted by conceptual focus, logical synthesis, as well as structure and linguistic clarity. It needs extensive revision before consideration for publication.

Major Comments

  1. The paper currently reads more like an encyclopedic collection or textbook chapter than a hypothesis-driven scientific review. Much of what it contains is peripheral or general to the main subject and does not meaningfully contribute to a distinct argument. Authors are strongly urged to state the major goal or key hypothesis in early portions of the introduction, develop an unambiguous conceptual framework to govern the discussion, and to avoid background information except where it is directly pertinent to rationale or therapeutic implication of PTH for neuromuscular interaction.

- We fixed our introduction part. Before, we had adjusted some background information following other reviewer comments and feedback. We hope the updated one could better emphasize our aim in this review article (1. Introduction, Line 28-45).

  1. Introduction

Parathyroid hormone (PTH) is a critical endocrine regulator of calcium and phosphate homeostasis, primarily known for its effects on bone and kidney metabolism [1]. Recent studies have highlighted its broader role in musculoskeletal health, particularly its effects on skeletal muscle. Meanwhile, bone and muscle are closely interconnected through both mechanical loading and biochemical signaling, playing key roles in muscle metabolism and function [2,3]. Furthermore, the nervous system is essential for maintaining muscle mass and function as neuromuscular signaling regulates muscle contraction, protein synthesis, and energy metabolism through calcium control [4,5]. Considering the classical role of PTH and the muscle symptoms (muscle weakness, myopathy, etc.) that occur in hyperparathyroidism and hypoparathyroidism, it is thought that PTH may affect these musculoskeletal and neuromuscular interactions in a complex manner [6]; however, there is a lack of comprehensive information on the systemic and local effects of PTH on skeletal muscle. Investigating the molecular pathways involved in PTH-mediated muscle regulation is expected to provide valuable insights into musculoskeletal diseases—particularly those related to aging, metabolic disorders, and endocrine dysfunction—and to suggest targeted treatment strategies for various musculoskeletal conditions.

Our review synthesizes emerging evidence on the non-classical roles of PTH in skeletal muscle and neuromuscular junction function. By integrating findings from clinical, in vivo, and in vitro studies, we provide a novel perspective on PTH as a potential therapeutic modulator in the muscle–bone–nerve crosstalk, addressing a critical gap in current musculoskeletal research.

  1. The skeletal muscle biology, hormone regulation, and bone-muscle-nerve physiology sections are informative but highly descriptive. These can be significantly condensed and reorganized to provide only essential background information for appreciation of novelty and scope of the review.

- We have designed additional tables 1 and 2, which summarize the study designs and results and the pathways, signalings, and mechanisms involved. We hope that these two summary tables can clarify and provide a concise overview of the contents we present in the relevant sections (Table 1: Line 232-233, Table 2: Line 293-294).

Table 1. Bone and skeletal muscle interactions.

Osteokines/Myokines

Effects on Bone

Mechanism/Signaling Involved

Effects on Skeletal Muscle

Mechanism/Signaling Involved

FGF23

In vitro: ↓ BMSCs osteogenesis, ↓ mature osteoblast mineralization [27]

FGFR3-ERK

In vitro: ↓ Muscle cell differentiation [30]

Insulin/IGF-1, klotho

Clinical: Serum level associated with low bone mass [28]

Relevant to skeletal muscle wasting [33]

PGE2

Regulating both bone resorption and formation processes [34,35]

In vitro: Myogenesis [38]

EP4 receptor

In vivo, in vitro: Muscle regeneration and strength [39]

Muscle-specific stem cells

Related: Wnt, β-catenin [36], cAMP/PKA [37]

Osteocalcin

Glucose metabolism, reproduction, and cognition [41]

In vitro: ↑ Glucose transport [43]

Clinical, in vivo: ↑ Muscle uptake [44,45]

In vitro: ↓ Osteoclasts differentiation [42]

GPRC6A

IGF-1

↑ Osteoblasts differentiation, bone production [47]

↑ Protein synthesis and regeneration

↓ Muscle atrophy [101]

Related: PI3K/Akt, MAPK/ERK [46]

Sclerostin

Clinical, in vivo, in vitro: ↓ Bone formation [50,51]

Wnt [49]

Clinical: ↑ Muscle weakness [54]

Myostatin

↓ Bone formation

Metabolism [57]

↓ Muscle mass [55]

In vitro: ↓ Osteoblastic differentiation [58]

Osteocyte-derived exosomal miR-218

↓ Osteoclast differentiation [59]

RANKL, NFATC1

Related: ERK1/2, Wnt, TGF-β1, IGF-1 [55]

Osteoactivin

In vivo, in vitro: ↓ Osteoclastogenesis [102]

CD44-ERK

In vivo: Protection from fibrosis [63]

MMP-3, MMP-9

In vivo, in vitro: ↑ Bone formation [66]

TGF-β

IL-6

↑ Bone growth [70,71]

↑ Formation, growth, regeneration, satellite-cell-dependent myogenesis [69]

↑ Bone loss (in several osteolytic diseases) [70,71]

↑ Protein synthesis and breakdown

Engaged with muscle atrophy [69]

IL-7

In vivo: ↑ Bone loss [74]

RANKL

Might affect satellite cells [72,73]

IL-15

Bidirectional regulatory function [79]

Clinical: ↑ Myoblast development, fiber growth; ↓ protein breakdown [78]

Irisin

In vivo: ↓ Bone loss [84]

In vitro: Mitochondrial biogenesis [82]

In vivo:

↑ Myogenesis [83]

↓ Muscle atrophy [84]

Related: MAPK [85], ERK/STAT, BMP/SMAD [87], Wnt/β-catenin [88]

FGF2

Bone growth [92–94]

Clinical, in vivo: ↑ Muscle growth, intramuscular adipogenesis [91]

miR-29a/SPARC

In vivo: Bone marrow MSC Osteogenesis

ERK/Runx2 [95]

Musclin

Bone resorption [100]

RANKL

Glucose metabolism [96,97]

In vivo: ↑ Physical endurance [98]

Mitochondrial biogenesis

BMSCs, bone marrow mesenchymal stem cells; GPRC6A, G protein-coupled receptor class C group 6; RANKL, receptor activator of nuclear factor κB ligand; NFATC1, nuclear factor of activated T-cells; MMP, matrix metalloprotease; MSCs, mesenchymal stem cells.

Table 2. Muscle and nerve communication.

Factors

Effects on Nerve

Mechanism/Signaling Involved

Effects on Skeletal Muscle

Mechanism/Signaling Involved

Motor neurons

Clinical: Affecting muscle fiber morphology and phenotype [118,119]

In vivo: Differentiation of slow muscles [122]; affects contractile speed of re-innervated muscle [123]

Gap junctions / NMJs

In vitro: ↑ Myoblast fusion [120]

Intercellular communication

Poor signal transmission and muscular weakening in aging [125,126]

NMJ deteriorates, mitochondria mechanism

Neural and hormonal influences

↑ Muscle development [121]

Isogenes

DOK7

In vivo: ↑ Muscles and motor activities [127]

↑ NMJ innervation

MuSCs

NMJ repair and maintenance [137–141]

Myofiber components, derived factors,

associated satellite cells

Muscle repair and regeneration [125,130]

BDNF

↑ Hippocampal neurons, neuronal plasticity, and synaptogenesis

↓ Neuroinflammation [132–134]

Supporting muscle regeneration and utilization [131]

Irisin

In vitro: Regulating astrocytes, neuroprotective effects [135]

Interleukins, COX-2, AKT, NFκB

In vitro: Mitochondrial biogenesis [82]

In vivo:

↑ Myogenesis [83]

↓ Muscle atrophy [84]

In vitro: Neural generation and development [136]

Post-neural progenitor formation

NMJs, neuromuscular junctions; MuSCs, muscle stem cells; BDNF, brain-derived neurotrophic factor.

  1. The section on the molecular biology of PTH and localization to receptors is too lengthy and just recounts familiar facts without strategic importance to the major objectives of the review. The latter ought to be focused on the functional importance of PTH in relation to bone-muscle-nerve signaling, emphasize recent advances and unresolved mechanistic insight, and minimize well-documented endocrine activities unless directly relevant to muscle/nerve crosstalk.

- We modified this section following your comment. We shortened the basic information, and emphasized recent advances and unresolved mechanistic insight (3. Parathyroid Hormone, Line 66-94).

  1. Parathyroid Hormone

PTH is the product of the parathyroid glands, nodular structures usually located along the dorsal part of the thyroid [7]. PTH, calcitriol (1,25-dihydroxyvitamin D), and fibroblast growth factor 23 (FGF23) are three key hormones modulating calcium and phosphate homeostasis. PTH synthesis occurs within the parathyroid glands, starting with a 115-amino-acid polypeptide; it then produces the primary hormone, active 84-amino-acid PTH, which is stored, secreted, and functions in the body [1]. All of the known biological actions operate within the 34 residues of its NH2-terminal (PTH 1–34) [1,8].

PTH receptors are classified as PTH1R, PTH2R, or nonclassical receptors. Research has revealed the expression of PTH1R in skeletal muscle, including satellite cells [9]. Parathyroid hormone-related protein (PTHrP) exhibits similarities to PTH, frequently acting in a local paracrine or autocrine manner [10]. Both PTH and PTHrP can be recognized by PTH1R because of the high degree of similarity in the amino-terminal regions of these two peptides [9,11].

PTH acts differently on the neuromuscular system depending on the degree of exposure. Intermittent PTH exposure, such as in osteoporosis treatments, has been shown to have anabolic effects on bone, which might indirectly benefit muscle by enhancing bone-derived signaling molecules and providing mechanical support [12]. Additionally, PTH can modulate neuromuscular function, potentially influencing the nerve conduction and synaptic activity essential for muscle contraction and coordination [13,14]. However, chronic high PTH levels—as seen in conditions such as hyperparathyroidism and chronic kidney disease—are associated with muscle weakness, increased protein degradation, and impaired muscle cell regeneration [6,15].

To the best of our knowledge, the United States Food and Drug Administration (FDA) has approved three PTH-based medicines for clinical use. Forteo (teriparatide), a recombinant PTH (1–34) fragment, is authorized for the treatment of osteoporosis in postmenopausal women and men at high risk for fractures. Tymlos (abaloparatide), a synthetic analog of PTH-related peptide (PTHrPs 1–34), is also indicated for the treatment of osteoporosis. Yorvipath (palopegteriparatide), approved in 2024, is a prodrug of PTH (1–34) intended to provide continuous PTH exposure for adults with hypoparathyroidism. These medicines are critical for treating bone and mineral metabolism problems, yet their emerging roles in skeletal muscle and neuromuscular health are not fully understood. These agents may exert direct or indirect effects on muscle regeneration and neuromuscular junction stability, representing a promising but mechanistically underexplored therapeutic avenue.

  1. Rather than enumerating facts within distinct subtopics, the authors can aim at conceptual integration. For example: compare the effects of intermittent versus chronic PTH exposure on neuromuscular health, integrate bone-derived and nerve-mediated mechanisms to display shared axes of signaling altered by PTH, include visual conceptual models (e.g., diagrams, summary tables) for targets for treatment, signal transduction pathways, or organ-to-organ communication.

- We found this suggestion very valuable, and a great idea. We have added additional content based on your comments. We hope that this information based on our discussion will provide further insight into the goals we want to present to our readers (Part 7.3, Line 462-485).

Chronic PTH elevation—as observed in primary or secondary hyperparathyroidism—leads to bone complications, muscle atrophy and weakness, and neuromuscular manifestations [186,226,227]. This persistent exposure status may disrupt calcium and phosphorus balance [186] and trigger endothelial dysfunction [216]. In contrast, intermittent administration of PTH, such as daily or every-other-day injections of teriparatide (PTH 1–34), has shown anabolic effects on the skeleton. This mode of delivery mimics the physiological pulsatility of PTH and is associated with increased bone functional outcomes in both animal models and clinical observations [187]. Likewise, many studies have shown the benefits of this method on the skeletal muscle, as we have summarized and discussed in the previous sections. These bases underscore the critical importance of exposure patterns: while chronic exposure is detrimental to musculoskeletal and neural health, intermittent PTH may be protective and pro-regenerative, supporting careful therapeutic titration in musculoskeletal disease contexts.

Recent advances suggest that organ-to-organ communication mediated by PTH signaling can be leveraged for therapeutic purposes. PTH acts as a central regulator at the interface of bone, muscles, and nerves, engaging shared molecular signaling axes that coordinate tissue function and regeneration. The involvement of these three tissues in key regulators such as Wnt [228,229], IGF-1/AKT [47,101,193], MAPK [230–232], and the calcineurin-NFAT pathway [233,234] highlights a multi-organ crosstalk mechanism. Targeting these shared pathways offers the potential to simultaneously modulate osteogenesis, myogenesis, and neuromuscular synaptic maintenance. Bone-, muscle-, and nerve-derived factors like IGF-1, irisin, and BDNF act systemically to promote musculoskeletal health and regeneration and support neuromuscular junction structure. Therapeutic strategies aimed at fine-tuning PTH delivery (e.g., pulsatile analogs) or modulating PTH-induced content may open new avenues for treating age-related musculoskeletal and NMJ degeneration, particularly in conditions like osteosarcopenia or sarcopenic neurodegeneration. The complex relationships between PTH, muscle, bone, and nerves are the subjects of ongoing investigations and have implications for aging, metabolic disorders, and musculoskeletal diseases.

  1. The discussion of PTH-based therapies is overly optimistic. The authors need to balance the discussion by including translational challenges such as: Delivery methods (e.g., instability of peptides, targeting tissues), Immunogenicity, Safety and off-target effects over the long-term, Clinical trial design difficulties in neuromuscular diseases

- We would like to add some lines to part 7.3 to discuss more about the challenges in PTH studies relevant to drug stability, targeting tissues, immunogenicity, safety… (Line 438-452)

Most of the preclinical studies have not verified the direct action of PTH on muscle via its receptors. Further research is needed to determine the specific effects of PTH in terms of the duration of action and effects of concentration. In clinical settings, nearly all published research has focused on comparing changes in PTH serum levels with skeletal muscle features; however, the results seem to indicate no significant connection. PTH is known for its instability; it degrades relatively quickly, requiring the careful handling of blood samples to ensure accurate measurements. Proper pre-analytical conditions, including specimen type, sampling time, and storage, are crucial for reliable PTH testing [217]. Furthermore, PTH has immunomodulatory effects, impacting various immune cells and functions, but the clinical significance remains unclear [218]. While PTH replacement medication, such as teriparatide, is usually considered safe and effective for treating hypoparathyroidism, it can have side effects (most notably, hypercalcemia) [219]. When used for fracture healing, PTH has a well-established safety profile, with mild side effects such as bruising at the injection site [220]. Evidence is still lacking on how PTH treatment can transform skeletal muscle characteristics, which is surprising given that several PTH products have already been FDA-approved. This leaves a gap for further investigations on the effects of PTH and its detailed mechanisms in humans.

  1. The mention of EV-derived miRNAs and other new modulators is intriguing but underdeveloped. Mention of these features should be prominent, contextually linked to PTH-induced nerve or muscle effects, and supported by recent high-impact references.

- The topic of EV-derived miRNAs is a promising trend. We would like to add some information and our discussion about future direction to part 7.3 (Line 453-461).

Recent studies have revealed that extracellular vesicles (EVs)—particularly exosomes—and their cargo of extracellular RNAs (exRNAs) are central players in the communication between skeletal muscle, bone, and the nervous system. These vesicles shuttle bioactive molecules—including regulatory microRNAs—that influence processes such as osteogenesis, synaptic maintenance, and myogenesis. Growing evidence suggests that muscle- and bone-derived EVs can cross-regulate each other’s tissues, shaping local regeneration and systemic homeostasis [221–223]. Importantly, PTH has emerged as a modulator of EV composition, potentially altering EV-derived miRNA profiles to influence bone turnover [145,224,225]. This evolving area offers a novel perspective beyond traditional hormone signaling, pointing to PTH-EV-miRNA axes as promising therapeutic targets in musculoskeletal and neuromuscular disorders.

  1. The conclusion is typically a summary of earlier content. It should declare primary mechanistic shortcomings in PTH signaling in bone, nerve, and muscle, outline experimental or clinical approaches to fill them, and indicate areas of highest priority for future therapeutic research.

- We modified the conclusion part. We hope this could better show the potential directions for future studies and development in this field (8. Conclusions and Future Directions, Line 494-500).

In this review, we highlight the potential to investigate the effects of PTH on osteokines, myokines, neuromuscular junctions, and other new modulators (e.g., EV-derived miRNAs) that may positively influence the musculoskeletal and neuromuscular axes. Clinical studies focused directly on skeletal muscle, which were previously limited, can now be designed to examine changes in the relevant organs. Future research should continue to explore how PTH-targeted therapies can optimize muscle function while minimizing potential adverse effects. By addressing these complex interactions, novel strategies can be developed to improve mobility, strength, and overall quality of life in affected individuals.

  1. The figures are unreadable and require an increased image resolution, larger font size, better labeling of molecular pathways and their relationship with PTH signaling.

- We increased the font size and image resolution for both Figure 1 and Figure 2. We presented how PTH acts on the osteokines, myokines, and the neuromuscular junction in part 7. However, because of the large amount of information, we could not provide them all in the figures. We hope our figures could share a general view of our whole review.

Figure 1. PTH acts on osteokines and myokines.

Figure 2. High Potential for PTH to enhance the NMJ and its components.

Additional Recommendations:

  1. Consider about dividing the review into mechanistic segments (e.g., PTH's role in bone-muscle, nerve-muscle, and bone-nerve axes) for thematic consistency.

- We have revised the manuscript for thematic consistency as below.

7.1. PTH's role in the bone–muscle axis (Line 340)

PTH, FGF23, and 1,25(OH)2 vitamin D (1,25D) are fundamental to calcium and phosphate homeostasis and together create acknowledged endocrinologic feedback loops [161,162]. PTH is also reported to upregulate FGF23 expression through the PKA and Wnt pathways [163].

Some authors have recognized that PTH stimulates PGE2 production [164–166]. Especially when cortisol levels are low, PGE2 released as a result of PTH effects is robust [165].

Several attempts have been made to clarify a link between PTH and osteocalcin. In 1997, Yu reported that activation of the osteocalcin promoter caused by PTH was rapid and regulated via the cAMP-dependent protein kinase A pathway [167]. Three years later, Boguslawski et al. extended these research methods to determine other pathways. That team claimed that both PKA-dependent and PKC-dependent mechanisms mediate the regulation of osteocalcin transcription [168]. PTH significantly raises osteocalcin mRNA levels in MC3T3-E1 pre-osteoblast cells and the primary cultures of bone marrow stromal cells via multiple signaling pathways that require OSE1 and associated nuclear proteins [169].

IGF-1 plays a crucial role in mediating PTH's anabolic effects on bone. As shown in both in vitro and in vivo experiments, IGF-1 mRNA and protein levels are upregulated by PTH [170–172]. Antibodies to IGF-1 prevent PTH and PTHrP from stimulating the formation of aggrecans in chondrocytes [173], and in in vitro experiments, PTH therapy has enhanced the number of cells in osteoblasts produced by IGF-1 knockout mice [174]. In the presence of IGF1R, PTH can act on bone formation [175] and activate osteoprogenitor cell proliferation and differentiation in mature osteoblasts [176]. Several signaling pathways involving IGF-1 and PTH have been shown to act in bone, including RANKL [176], ephrin B2/EphB4 [177], and PTH/Indian hedgehog [178].

The SOST gene encodes sclerostin and is inhibited by the intermittent or continuous administration of PTH in vitro and in vivo. Possible pathways and factors involved include the cAMP/PKA pathway and the proteasomal degradation of Runx2 [179,180]. After menopause, intermittent PTH treatment significantly downregulates serum sclerostin levels [181]. These data demonstrate that PTH stimulation decreases the production and secretion of sclerostin.

Data from multiple sources have identified a connection between PTH and IL-6. For example, PTH modulation was found to be responsible for the production of IL-6 in vitro and in vivo. Clinically, IL-6 serum levels are increased in patients with primary hyperparathyroidism [182–184].

Few studies have analyzed the relationship between PTH and myostatin. In fact, we could not find any in vitro or in vivo evidence concerning this relationship. In the STRAMBO study, Szulc found that the PTH serum level was not associated with myostatin concentrations [185]. Martino reported an indirect comparison in which the circulating levels of PTH and myostatin were negatively associated with the maximum voluntary contraction [186]. Further research is needed to clarify the connection between these factors.

Irisin is a novel myokine and adipokine that has attracted great attention lately. Studies have found that several metabolic actions of PTH are apparently opposed to those of irisin [187,188]. Palermo’s in vitro preclinical study reported the only findings about direct biological crosstalk between PTH and irisin [147]. Therefore, a new approach is needed to clarify the interactions between these two hormones.

Several studies have examined how PTH works with FGF2. In osteoporotic patients treated with PTH, serum FGF2 increases, and in osteoblasts, PTH elevates FGF2 mRNA and protein expression [189–191]. Additionally, although the osteoclastogenic effects of PTH are diminished in FGF2-null mice, the osteoclast-activating effect of PTH re-emerges in cells cocultured with osteoblasts or treated with FGF2. Therefore, PTH appears to increase osteoclast formation and bone resorption in mice, partly through endogenous FGF2 synthesis by osteoblasts [192].

Until recently, no reliable evidence has demonstrated any interactions between PTH and osteoactivin, IL-7/IL-15, or musclin.

7.2. PTH's role in the nerve–muscle axis (Line 392)

Several biological agents and drugs exert dual effects on both skeletal muscle and nerve tissues, making them promising candidates for treating neuromuscular disorders. For instance, IGF-1 enhances muscle in both anabolic and catabolic pathways [48] and plays a significant role in nerve health and regeneration [193]. BDNF and GDNF (glial cell line-derived neurotrophic factor) are neurotrophic factors that stabilize neuromuscular junctions, promote neural health, and target skeletal muscle [194,195]. Testosterone and β2-adrenergic agonists contribute to muscle hypertrophy and nerve activities [196–199]. Additionally, exercise-induced myokines such as IL-6 and irisin mediate communication between muscles and nerves, influencing both regeneration and synaptic maintenance [200–203]. These molecules highlight the interconnected nature of muscle and nerve systems and are being explored for therapies targeting aging, sarcopenia, injury, and neurodegenerative diseases.

Calcium plays a crucial role in the function of NMJs: it is essential for neurotransmitter release, regulates muscle contraction, and maintains neuromuscular function [204–206]. Without Ca2+, ACh is not released and muscles do not contract. Systemic calcium imbalances, such as hypocalcemia or hypercalcemia, can thus negatively affect NMJs [207–209]. These clinical conditions might also be associated with PTH disorders (hypoparathyroidism or hyperparathyroidism). Abnormal MuSK expression, AChR clustering, and nerve branching can be caused by Ca2+ signal loss [206]. Lack of Ca2+ can also cause variations in NMJ structures and functions and induce a sustained stress response in muscles [210].

An in vitro study showed that PTH could boost the mean speed of both anterograde and retrograde organelle traffic on axons [211]. PTH (1–34) treatment can affect axonal regeneration by enhancing endogenous BMP-7 in rat Schwann cells [212]. Moreover, neurons in the subfornical organ—which lies above the third ventricle and modulates body fluid homeostasis—can be activated by circulating PTH [213].

ACh is a neurotransmitter that performs numerous roles in the brain and other organ systems and is commonly associated with the NMJ. Some authors have suggested a relationship between PTH and ACh. In the rat superior cervical ganglion (evaluated in vitro), ACh is released when PTH increases and calcitonin is inhibited [214]. The effect of PTH on 3H-acetylcholine synthesis in rat parathyroid glands has been investigated in vitro, and it was found to inhibit cholinergic activity [215]. Furthermore, ACh can be preserved by PTH-induced oxidative stress [216].

As noted, agrin, rapsyn, LRP4, and MuSK are important parts of the NMJ. LRP4 can also be involved in Wnt signaling activity, which can influence skeletal muscle. Nevertheless, little research has directly linked PTH with these NMJ components.

  1. Insert a table summarizing comparison of experimental models (in vitro, in vivo, clinical) and their principal findings on PTH's neuromuscular actions.

- We tried our best to perform the searches about PTH’s direct actions on the neuromuscular junction and its components. The results were mostly the effect of PTH on the axons and acetylcholine activities, as we presented in part 7.2. We would like to present them in the Table 4 (Line 423-424).

Table 4. PTH’s actions on NMJ components.

Target of NMJ components

Study design

PTH’s effects

Ref.

Axon / Neuron

In vitro

PTH boosts the mean speed of both anterograde and retrograde organelle traffic on axons

[211]

In vivo

PTH (1–34) treatment can affect axonal regeneration by enhancing endogenous BMP-7 in rat Schwann cells

[212]

In vivo

Circulating PTH activates neurons in the subfornical organ

[213]

Acetylcholine

activities

In vitro

In the rat superior cervical ganglion, ACh is released when PTH increases and calcitonin is inhibited

[214]

In vitro

PTH affects 3H-acetylcholine synthesis in rat parathyroid glands

[215]

In vitro

PTH-induced oxidative stress preserves ACh

[216]

PTH, parathyroid hormone; BMP, bone morphogenetic protein; ACh, acetylcholine.

  1. Describe if muscle-specific or nerve-specific administration of PTH analogs has been examined.

- We added a paragraph to part 7.2 (Line 393-402) to describe if muscle-specific or nerve-specific administration of PTH analogs has been examined.

Several biological agents and drugs exert dual effects on both skeletal muscle and nerve tissues, making them promising candidates for treating neuromuscular disorders. For instance, IGF-1 enhances muscle in both anabolic and catabolic pathways [48] and plays a significant role in nerve health and regeneration [193]. BDNF and GDNF (glial cell line-derived neurotrophic factor) are neurotrophic factors that stabilize neuromuscular junctions, promote neural health, and target skeletal muscle [194,195]. Testosterone and β2-adrenergic agonists contribute to muscle hypertrophy and nerve activities [196–199]. Additionally, exercise-induced myokines such as IL-6 and irisin mediate communication between muscles and nerves, influencing both regeneration and synaptic maintenance [200–203]. These molecules highlight the interconnected nature of muscle and nerve systems and are being explored for therapies targeting aging, sarcopenia, injury, and neurodegenerative diseases.

We appreciate your comments and feedback. We strive to improve our reviews as much as possible and provide a constructive perspective for future research. We are grateful very much.

Round 2

Reviewer 1 Report (Previous Reviewer 2)

Comments and Suggestions for Authors

Thank you for all your work and effort to address the previous comments.

I believe this revised version represents a significant and favorable evolution in the content of your review. The overall quality and depth of the manuscript have noticeably improved.

The inclusion of Tables 1, 2, 3 (which was previously Table 1), and 4 is a strong addition. These tables provide a comprehensive and easily interpretable overview of the concepts presented, which greatly facilitates the reader's understanding of the complex interactions discussed.

Furthermore, I note that the reference list has been updated. The removal of some references and the addition of more recent ones have resulted in a more current and extensive bibliography. This undoubtedly contributes to a more thorough and up-to-date analysis of the information included in the review, as is evident from the changes made to the main text.

Regarding minor points that could further enhance the manuscript:

While the tables are excellent, some information presented in Tables 1 and 2 currently lacks direct supporting references. Please ensure that all data and statements within these tables are adequately referenced or clarify if they are general knowledge not requiring citation.

Tables appear to have a larger font size compared to the main text, which affects their visual integration. Please adjust the formatting to ensure a consistent font size throughout the manuscript for all tables and figures.

Although the methodology section mentions a new search in the various databases, there isn't explicit evidence presented within the manuscript that this new search was indeed conducted. While the updated reference list strongly suggests an effective literature review, providing a brief statement confirming the scope or outcome of this new search, if applicable, could further strengthen this section.

Overall, this is a strong manuscript, and the revisions have significantly enhanced its quality.

Author Response

Thank you for all your work and effort to address the previous comments.

I believe this revised version represents a significant and favorable evolution in the content of your review. The overall quality and depth of the manuscript have noticeably improved.

The inclusion of Tables 1, 2, 3 (which was previously Table 1), and 4 is a strong addition. These tables provide a comprehensive and easily interpretable overview of the concepts presented, which greatly facilitates the reader's understanding of the complex interactions discussed.

Furthermore, I note that the reference list has been updated. The removal of some references and the addition of more recent ones have resulted in a more current and extensive bibliography. This undoubtedly contributes to a more thorough and up-to-date analysis of the information included in the review, as is evident from the changes made to the main text.

Regarding minor points that could further enhance the manuscript:

While the tables are excellent, some information presented in Tables 1 and 2 currently lacks direct supporting references. Please ensure that all data and statements within these tables are adequately referenced or clarify if they are general knowledge not requiring citation.

We have added direct references to the content. Previously, the content in each single line displays the information for 1 organ (bone, muscle, or nerve); the related mechanisms have the same citation as the effect, so we only leave the reference in 1 of the 2 columns. We have added it to avoid confusion for the reader (Table 1: Line 271-272, Table 2: Line 341-342).

Table 1. Bone and skeletal muscle interactions.

Osteokines/Myokines

Effects on Bone

Mechanism/Signaling Involved

Effects on Skeletal Muscle

Mechanism/Signaling Involved

FGF23

In vitro: ↓ BMSCs osteogenesis, ↓ mature osteoblast mineralization [27]

FGFR3-ERK [27]

In vitro: ↓ Muscle cell differentiation [30]

Insulin/IGF-1, klotho [30]

Clinical: Serum level associated with low bone mass [28]

Relevant to skeletal muscle wasting [33]

PGE2

Regulating both bone resorption and formation processes [34,35]

In vitro: Myogenesis [38]

EP4 receptor [38]

In vivo, in vitro: Muscle regeneration and strength [39]

Muscle-specific stem cells [39]

Related: Wnt, β-catenin [36], cAMP/PKA [37]

Osteocalcin

Glucose metabolism, reproduction, and cognition [41]

In vitro: ↑ Glucose transport [43]

Clinical, in vivo: ↑ Muscle uptake [44,45]

In vitro: ↓ Osteoclasts differentiation [42]

GPRC6A [42]

IGF-1

↑ Osteoblasts differentiation, bone production [47]

↑ Protein synthesis and regeneration

↓ Muscle atrophy [101]

Related: PI3K/Akt, MAPK/ERK [46]

Sclerostin

Clinical, in vivo, in vitro: ↓ Bone formation [50,51]

Wnt [49]

Clinical: ↑ Muscle weakness [54]

Myostatin

↓ Bone formation

Metabolism [57]

↓ Muscle mass [55]

In vitro: ↓ Osteoblastic differentiation [58]

Osteocyte-derived exosomal miR-218 [58]

↓ Osteoclast differentiation [59]

RANKL, NFATC1 [59]

Related: ERK1/2, Wnt, TGF-β1, IGF-1 [55]

Osteoactivin

In vivo, in vitro: ↓ Osteoclastogenesis [102]

CD44-ERK [102]

In vivo: Protection from fibrosis [63]

MMP-3, MMP-9 [63]

In vivo, in vitro: ↑ Bone formation [66]

TGF-β [66]

IL-6

↑ Bone growth [70,71]

↑ Formation, growth, regeneration, satellite-cell-dependent myogenesis [69]

↑ Bone loss (in several osteolytic diseases) [70,71]

↑ Protein synthesis and breakdown

Engaged with muscle atrophy [69]

IL-7

In vivo: ↑ Bone loss [74]

RANKL

Might affect satellite cells [72,73]

IL-15

Bidirectional regulatory function [79]

Clinical: ↑ Myoblast development, fiber growth; ↓ protein breakdown [78]

Irisin

In vivo: ↓ Bone loss [84]

In vitro: Mitochondrial biogenesis [82]

In vivo:

↑ Myogenesis [83]

↓ Muscle atrophy [84]

Related: MAPK [85], ERK/STAT, BMP/SMAD [87], Wnt/β-catenin [88]

FGF2

Bone growth [92–94]

Clinical, in vivo: ↑ Muscle growth, intramuscular adipogenesis [91]

miR-29a/SPARC [91]

In vivo: Bone marrow MSC Osteogenesis [95]

ERK/Runx2 [95]

Musclin

Bone resorption [100]

RANKL [100]

Glucose metabolism [96,97]

In vivo: ↑ Physical endurance [98]

Mitochondrial biogenesis

BMSCs, bone marrow mesenchymal stem cells; GPRC6A, G protein-coupled receptor class C group 6; RANKL, receptor activator of nuclear factor κB ligand; NFATC1, nuclear factor of activated T-cells; MMP, matrix metalloprotease; MSCs, mesenchymal stem cells.

Table 2. Muscle and nerve communication.

Factors

Effects on Nerve

Mechanism/Signaling Involved

Effects on Skeletal Muscle

Mechanism/Signaling Involved

Motor neurons

Clinical: Affecting muscle fiber morphology and phenotype [118,119]

In vivo: Differentiation of slow muscles [122]; affects contractile speed of re-innervated muscle [123]

Gap junctions/NMJs

In vitro: ↑ Myoblast fusion [120]

Intercellular communication [120]

Poor signal transmission and muscular weakening in aging [125,126]

NMJ deteriorates, mitochondria mechanism [125,126]

Neural and hormonal influences

↑ Muscle development [121]

Isogenes [121]

DOK7

In vivo: ↑ Muscles and motor activities [127]

↑ NMJ innervation [127]

MuSCs

NMJ repair and maintenance [137–141]

Myofiber components, derived factors,

associated satellite cells [137–141]

Muscle repair and regeneration [125,130]

BDNF

↑ Hippocampal neurons, neuronal plasticity, and synaptogenesis

↓ Neuroinflammation [132–134]

Supporting muscle regeneration and utilization [131]

Irisin

In vitro: Regulating astrocytes, neuroprotective effects [135]

Interleukins, COX-2, AKT, NFκB [135]

In vitro: Mitochondrial biogenesis [82]

In vivo:

↑ Myogenesis [83]

↓ Muscle atrophy [84]

In vitro: Neural generation and development [136]

Post-neural progenitor formation [136]

NMJs, neuromuscular junctions; MuSCs, muscle stem cells; BDNF, brain-derived neurotrophic factor.

Tables appear to have a larger font size compared to the main text, which affects their visual integration. Please adjust the formatting to ensure a consistent font size throughout the manuscript for all tables and figures.

We have adjusted the font size in the tables to match the font size of the main text (Table 1: Line 271-272, Table 2: Line 341-342, Table 3: Line 348-349, Table 4: Line 493-494).

Although the methodology section mentions a new search in the various databases, there isn't explicit evidence presented within the manuscript that this new search was indeed conducted. While the updated reference list strongly suggests an effective literature review, providing a brief statement confirming the scope or outcome of this new search, if applicable, could further strengthen this section.

We would like to modify our methodology part as below (Line 48-77):

  1. Methodology

A comprehensive literature search was conducted to identify relevant studies published up to 2025, with a focus on the role of parathyroid hormone (PTH) in skeletal muscle and its interactions with bone and nerve systems. The primary database used was PubMed, supplemented by Scopus, Web of Science, ScienceDirect, and Embase to ensure broad coverage of biomedical and interdisciplinary research.

The following keywords and their combinations were used in the search: parathyroid hormone, PTH, skeletal muscle, hormones, bone–muscle, nerve–muscle, myokines, osteokines, neuromuscular junction, and NMJ. In addition, we included the names of specific myokines and osteokines (e.g., IL-6, irisin, FGF23, and osteocalcin) and NMJ components (e.g., acetylcholine cluster, Agrin, and MuSK) to refine and deepen the search. Operators such as "and" and "to" were applied to structure complex queries and optimize retrieval.

The inclusion criteria were as follows:

- Original research articles, reviews, and meta-analyses published in English.

- Studies addressing the role or effects of PTH (or its analogs) on skeletal muscle, neuromuscular junctions, or crosstalk between muscle and bone or nerve.

- Studies involving in vivo, in vitro, or clinical investigations.

The exclusion criteria were non-peer-reviewed sources, editorials, and conference abstracts without primary data.

To ensure the depth and accuracy of the review, reference lists of key publications were also manually examined to identify additional pertinent studies. All retrieved records were screened based on title and abstract, and a full-text review was conducted to confirm eligibility.

For example, to explore the effects of PTH on a myokine such as irisin and its target organ, skeletal muscle, we searched PubMed with the keywords "PTH and irisin and muscle" and found 7 results. Next, the keywords "irisin and muscle and bone" or "irisin and muscle and nerve" were searched to learn more about how this myokine affects these organs. The results were filtered to be relevant to the topic of our review. We repeated this process for the other databases. The most recent findings or reviews were prioritized to avoid duplicating earlier studies and to reflect the latest advancements.

Overall, this is a strong manuscript, and the revisions have significantly enhanced its quality.

We really appreciate your feedback. Your comments help us build and improve our review a lot. We sincerely thank you.

Reviewer 2 Report (New Reviewer)

Comments and Suggestions for Authors

In the revised article, the authors effectively responded to the reviewer's comments by making changes to the manuscript and figures, as well as providing thorough answers to all inquiries.

Author Response

In the revised article, the authors effectively responded to the reviewer's comments by making changes to the manuscript and figures, as well as providing thorough answers to all inquiries.

We really appreciate your feedback. Your comments help us build and improve our review a lot. We sincerely thank you.

This manuscript is a resubmission of an earlier submission. The following is a list of the peer review reports and author responses from that submission.

Round 1

Reviewer 1 Report

Comments and Suggestions for Authors

The paper deals with the Parathyroid Hormone as a Modulator of Skeletal Muscle.

The paper is a review but the content does not justify a publication as the authors ddid not enter in detailed description of the hormones involved in modulating skeletal muscle activity

I don't find enough merit in the paper

Comments on the Quality of English Language

the English style is quite good and this is not the main criticism of the article

Reviewer 2 Report

Comments and Suggestions for Authors

The submitted manuscript explores the role of Parathyroid Hormone (PTH) in skeletal muscle, a topic of considerable interest. The manuscript touches upon important interactions between bone, muscle, and nerves. While the theme is promising, several aspects require clarification and revision to strengthen the paper for potential publication. My comments are intended to be constructive and are outlined below, starting with major considerations followed by more specific points.

Major Comments

Clarity of Objective and Novel Contribution: The manuscript's specific objective and its intended novel contribution to the existing literature are not immediately clear. Could the authors explicitly state the primary goal of this review and highlight what new perspective or synthesis it offers beyond previous work? A clearer focus would help frame the presented information more effectively.

Methodology for Literature Search: As this is presented as a review article, transparency regarding the literature search and selection process is crucial. The review would be significantly strengthened by including details about the methodology used (e.g., databases searched, search terms and combinations, inclusion/exclusion criteria). This allows readers to understand the scope of the review and assess the potential risk of overlooking relevant studies.

Depth and Organization of Content: Several sections seem somewhat underdeveloped or presented in a way that could be confusing for the reader. There appears to be a mix of information from different study types (animal models, cell studies, human data) and contexts (e.g., exercise, various pathologies) without always clearly distinguishing between them or providing sufficient depth. A more thorough exploration of the topics and a potential reorganization of the material could enhance clarity and reader engagement. For instance, distinguishing clearly between in vitro/animal findings and human physiological/pathological data is essential throughout the text (see specific comments below).

Timeliness of References: While the references cited are generally relevant, a notable portion (approximately 60%) are more than ten years old. Although citing foundational studies is often necessary, the review would benefit from incorporating more recent findings to reflect the latest advancements in this rapidly evolving field. An updated literature search could enhance the manuscript's currency and impact.

Minor Comments and Specific Points

Language and Style: Careful revision of the text is recommended to ensure precise scientific language and consistent verb tense usage. Some terms used might be too informal or imprecise for a scientific publication (e.g., "a little bit," "not greatly related" - see comments on Lines 299, 310).

Acronyms: Please ensure all acronyms are defined upon their first use in the text, and potentially again in figure/table legends if appropriate. This includes acronyms within tables and figures themselves (e.g., Table 1, Figures 1 and 2).

Specific Content Clarifications (as examples):

Line 64: The statement "Current research reveals the expression of PTH1R in skeletal muscle, including satellite cells" should be supported by a specific citation.

Introduction of PTHrP: PTHrP is mentioned relatively early without a formal introduction outlining its origin or primary functions in the body, which might be helpful context for readers less familiar with it.

Lines 68-70: This sentence seems somewhat redundant, as the presence of a receptor generally implies the potential for the ligand to act on that cell/tissue. Consider rephrasing for conciseness or added value.

Lines 83-84: Why specifically mention sarcopenia and muscular osteoporosis? Could the rationale be provided, or should other related conditions like primary osteoporosis or immobility syndrome also be considered in this context?

Line 104: The phrase "Multiple lines of evidence indicate that..." is currently supported by only one reference (37). Please provide the additional evidence or rephrase to accurately reflect the supporting citations.

Section Titles (3.1, 3.2, 4.2, 4.3): The titles for sections 3.1 and 3.2 seem somewhat generic. These sections primarily list molecules; a more systematic presentation detailing their functions, mechanisms, context of production (physiological states, excess/deficit), perhaps in a table or revised text structure, could improve clarity. Similarly, the titles for 4.2 and 4.3 could be more descriptive of their specific content.

Level of Detail: Some information is presented very concisely, assuming significant background knowledge from the reader. Providing slightly more detail or explanation, particularly regarding the relevance of mentioned pathways or molecules within the context of the review's theme, would be beneficial. Examples include:

Lines 110-112: What is the specific relevance of PGE2 acting via the cited signaling pathways in the context of bone/nerve-muscle interactions mediated by PTH?

Line 122: Please define or briefly explain "GPRC6A" upon first use.

Lines 289-290: The text mentions PTH actions; what are the functional consequences of these specific actions in the described context?

Line 293: "dozens of weeks" is imprecise. Could a more specific timeframe or range be provided based on the cited literature?

Lines 299 and 310: Terms like "a little bit" (Line 299) and "...are not greatly related" (Line 310) are imprecise. Please use more quantitative or specific scientific language.

Tables and Figures:

Table 1: Needs a legend or footnote defining all acronyms used within the table.

Figures 1 and 2:

These figures are not currently cited or explicitly discussed in the main text. They should be integrated and explained.

The font size within the figures appears too small for easy readability.

A legend is required to explain all acronyms used.

The meaning of different line types (e.g., dashed vs. solid arrows, blunt-ended lines) needs to be clearly defined in the figure legend.

In summary, this manuscript addresses an interesting and relevant topic. However, significant revisions focusing on clarifying the review's objective and contribution, detailing the methodology, enhancing the depth and organization of content, updating the references, and addressing the specific points listed above are needed. I believe that addressing these concerns will substantially improve the manuscript's clarity, rigor, and potential impact.

I hope these comments are helpful for revising your work.